# Visualization of stem cell activity in pancreatic cancer expansion by direct lineage tracing with live imaging

**Takahisa Maruno[1], Akihisa Fukuda[1]\*, Norihiro Goto[1], Motoyuki Tsuda[1], Kozo Ikuta[1], Yukiko Hiramatsu[1], Satoshi Ogawa[1], Yuki Nakanishi[1], Yuichi Yamaga[1], Takuto Yoshioka[1], Kyoichi Takaori[2], Shinji Uemoto[2], Dieter Saur[3,4], Tsutomu Chiba[1,5], Hiroshi Seno[1]\***

[1]Department of Gastroenterology and Hepatology, Kyoto University Graduate School of Medicine, Kyoto, Japan; [2]Division of Hepatobiliary-Pancreatic Surgery and Transplantation, Department of Surgery, Kyoto University Graduate School of Medicine, Kyoto, Japan; [3]Department of Internal Medicine II, Klinikum rechts der Isar Technische Universität München, München, Germany; [4]Division of Translational Cancer Research, German Cancer Research Center (DKFZ) and German Cancer Consortium (DKTK), Heidelberg, Germany; [5]Kansai Electric Power Hospital, Fukushima-ku Osaka-shi, Osaka, Japan

**Abstract** Pancreatic ductal adenocarcinoma (PDAC) is a devastating disease. Although rigorous efforts identified the presence of 'cancer stem cells (CSCs)' in PDAC and molecular markers for them, stem cell dynamics in vivo have not been clearly demonstrated. Here we focused on Doublecortin-like kinase 1 (Dclk1), known as a CSC marker of PDAC. Using genetic lineage tracing with a dual-recombinase system and live imaging, we showed that Dclk1$^+$ tumor cells continuously provided progeny cells within pancreatic intraepithelial neoplasia, primary and metastatic PDAC, and PDAC-derived spheroids in vivo and in vitro. Furthermore, genes associated with CSC and epithelial mesenchymal transition were enriched in mouse Dclk1$^+$ and human DCLK1-high PDAC cells. Thus, we provided direct functional evidence for the stem cell activity of Dclk1$^+$ cells in vivo, revealing the essential roles of Dclk1$^+$ cells in expansion of pancreatic neoplasia in all progressive stages.

**\*For correspondence:**
fukuda26@kuhp.kyoto-u.ac.jp
(AF);
seno@kuhp.kyoto-u.ac.jp (HS)

**Competing interests:** The authors declare that no competing interests exist.

## Introduction

Pancreatic ductal adenocarcinoma (PDAC) has one of the worst prognoses among all human malignancies (*Siegel et al., 2016*). Therefore, understanding the biology of this lethal disease is urgently needed to develop the novel therapeutic approaches for it. For this purpose, one of the most critical questions in both cancer research and clinic is how PDAC is maintained and expanded after it has emerged. Recently, cancer stem cells (CSCs) have been considered as a subpopulation of cancer cells capable of self-renewing and producing progeny cells that are critical for cancer growth (*Al-Hajj et al., 2003*; *Bonnet and Dick, 1997*; *Lapidot et al., 1994*). This mechanism may underlie the maintenance of cancer and its resistance to conventional therapies. However, it remains unclear how such CSCs behave to expand their clones in the progression of tumor cells from preinvasive precursor lesions to invasive PDAC and aggressive metastatic lesions.

Recently, several studies focusing on Doublecortin-like kinase 1 (Dclk1) in PDAC and its precursor lesions, pancreatic intraepithelial neoplasia (PanIN), have been reported. Dclk1 is expressed by a small subset of PanIN and PDAC cells (*Bailey et al., 2014*; *Delgiorno et al., 2014*; *Ito et al., 2016*). Dclk1$^+$ cells in the normal pancreas harbor the potential function to initiate premalignant lesions

when the oncogenic *Kras* mutation is introduced together with inflammatory stimuli such as caerulein-induced pancreatitis in vivo (*Westphalen et al., 2016*). Another group showed a spheroid-forming capacity of acetylated tubulin$^+$/DCLK1$^+$ PDAC cells in vitro (*Bailey et al., 2014*). Although these reports demonstrate the potential of Dclk1$^+$ cells to form pancreatic tumors under specific conditions, the stem cell dynamics in vivo within established tumors has not been explored.

Lineage tracing assay is one of the most rigorous and powerful methods to clarify stem cell behavior and to define stem cell populations (*Kretzschmar and Watt, 2012*). However, so far, no longitudinal in vivo tracking of progeny of pancreatic tumor/CSCs for the maintenance of PDAC and its preinvasive precursor lesions has been reported in part due to technical issues. Therefore, the role of Dclk1$^+$ cells for the maintenance of PanIN and PDAC remains elusive. Furthermore, live imaging of expansion of tumor cells within same tumors in same mice has not yet been developed, mainly because it has also been technically challenging. In this context, to uncover the mechanisms of pancreatic tumor maintenance in vivo, we developed a novel live imaging system to robustly follow the dynamics of tumor cell expansion from CSCs by utilizing an inducible dual-recombinase system that combined flippase-*FRT* and Cre-*loxP* recombinations (*Schönhuber et al., 2014*) and two-photon excitation fluorescence imaging in mice. This study clearly demonstrated for the first time the pivotal stem cell activity of Dclk1$^+$ tumor cells for the maintenance and expansion of primary PDAC and its precursor lesions as well as metastatic lesions in vivo.

## Results

### Dclk1$^+$ cells in PanINs and PDACs also expressed CSC markers

We first investigated the proportion of Dclk1$^+$ cells in PanINs and PDACs established in *Pdx1-Flp; Kras$^{FSF-G12D/+}$* (KF) and *Pdx1-Flp; Kras$^{FSF-G12D/+}$; Trp53$^{frt/frt}$* (KPF) mice, well-established mouse models of pancreatic tumors (*Schönhuber et al., 2014*). PanINs that were positive for Alcian blue and Krt19 staining were developed in KF mice within 3 months of age (*Figure 1A–C*). We observed Dclk1 expression in a small subset of PanIN cells in KF mice (6.54 ± 1.32%, *Figure 1D*, *Figure 1—source data 1*). In KPF mice, PDACs that were also strongly positive for Krt19 were developed within 8 weeks of age (*Figure 1E and F*). Dclk1 staining was also detected only in a fraction of PDAC cells in this model (0.173 ± 0.029%, *Figure 1G*, *Figure 1—source data 2*). Similar to mouse PDACs, human PDACs displayed DCLK1 expression only in a limited proportion of PDAC cells (0.097 ± 0.026%, *Figure 1H*, *Figure 1—source data 3*).

We next examined whether mouse Dclk1$^+$ PanIN and PDAC cells expressed previously described CSC markers, including Epcam, Cd44, Cd24 (*Lee et al., 2008*; *Li et al., 2007*), and Aldh1a1 (*Kim et al., 2011*; *Rasheed et al., 2010*). In good agreement with previous reports showing that the single expression of Epcam, Cd44 or Cd24 alone is not restricted to CSC in PDAC (*Kure et al., 2012*), a relatively large proportion of PanIN and PDAC cells showed positive expression of Epcam, Cd44 and Cd24, whereas Aldh1a1 was expressed in a small subset of these cells (*Figure 1I–J*, *Figure 1—source data 4* and *5*). We observed colocalization between these CSC markers and Dclk1 in PanIN and PDAC (*Figure 1K*). These data suggested that Dclk1$^+$ tumor cells harbor the potential stemness in formed PanIN and PDAC.

### Dclk1$^+$ PanIN cells maintained mouse PanINs

To study the potential stem cell function of Dclk1$^+$ tumor cells, we applied lineage tracing method and examined whether Dclk1$^+$ tumor cells supply the progeny tumor cells within already established mouse PanINs. So far, most researchers have combined either *Pdx1-Cre* or *Ptf1a-Cre* mouse lines with *Kras$^{LSL-G12D}$* mouse to develop pancreatic tumors. However, with these mouse models, we were unable to perform lineage tracing of Dclk1$^+$ tumor cells, because the Cre-*loxP* system is already applied to activate oncogenic *Kras* and because *Rosa*-reporter has to be activated in another system. In order to solve this technical issue, we generated *Dclk1$^{CreERT2-IRES-EGFP/+}$; Rosa26$^{mTmG/+}$; Pdx1-Flp; Kras$^{FSF-G12D/+}$* (DRKF) mice (*Figure 2A*). In this mouse model, PanINs are spontaneously developed, thanks to the flippase-*FRT* recombination, which induces oncogenic *Kras$^{G12D}$* in Pdx1$^+$ pancreatic progenitor cells (*Figure 2A*). Further activation of Cre-*loxP* system with tamoxifen administration enables to trace Dclk1-lineages as EGFP$^+$ cells by switching Tomato to EGFP in potential descendent cells of Dclk1$^+$ cells in already formed PanINs (*Figure 2A and B*). In DRKF mice before

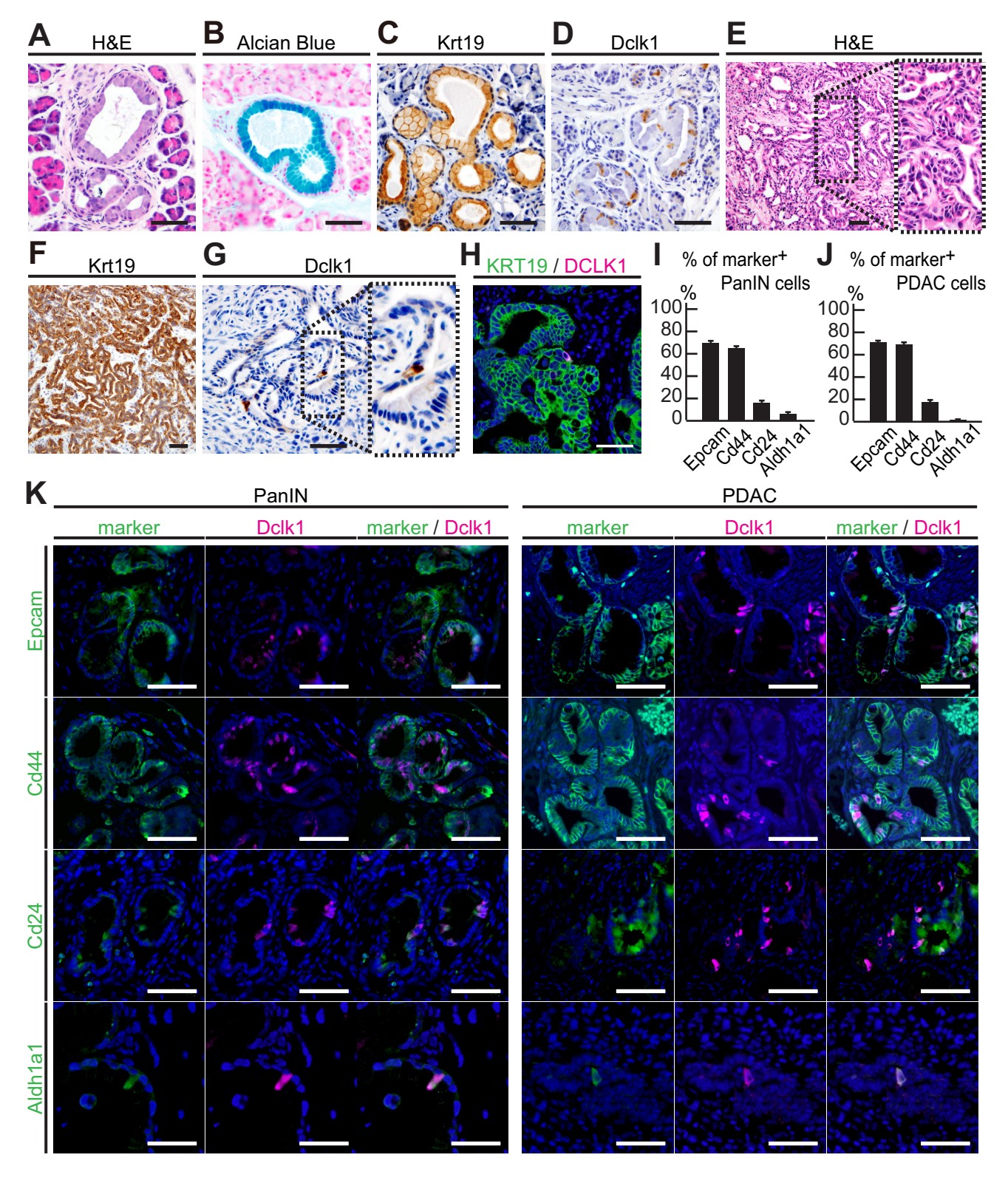

**Figure 1.** Dclk1+ cells presented in PanINs and pancreatic ductal adenocarcinomas (PDACs) also expressed cancer stem cell (CSC) markers. (A–D) Histological analysis of PanINs developed in 3-month-old *Pdx1-Flp; Kras^{FSF-G12D/+}* (KF) mice. (A) Hematoxylin and Eosin (H and E) staining. (B) Alcian Blue staining. (C) Immunostaining for Krt19. (D) Immunostaining for Dclk1 (mean ± SEM, *n* = 7, *n*: number of mice). Scale bar, 50 μm. (E–G) Histological analysis of PDACs developed in 8-week-old *Pdx1-Flp; Kras^{FSF-G12D/+}; Trp53^{frt/frt}* (KPF) mice. (E) Hematoxylin and Eosin staining. (F) Immunostaining for

*Figure 1 continued on next page*

Figure 1 continued

Krt19. (G) Immunostaining for Dclk1 (mean ± SEM, *n* = 8, *n*: number of mice). Scale bar, 50 μm. (H) Immunofluorescence staining for DCLK1 (magenta), Krt19 (green) and Hoechst (blue) in resected human PDACs (mean ± SEM, *n* = 7, *n*: number of case). Scale bar, 50 μm. (I) Quantification of the marker[+] cells in PanINs formed in KF mice (mean ± SEM; Epcam; *n* = 5, Cd44; *n* = 5, Cd24; *n* = 5, Aldh1a1; *n* = 5, *n*: number of mice). (J) Quantification of the marker[+] cells in PDACs formed in KPF mice (mean ± SEM; Epcam; *n* = 5, Cd44; *n* = 5, Cd24; *n* = 5, Aldh1a1; *n* = 5, *n*: number of mice). (K) Immunofluorescence staining for Dclk1 (magenta), Hoechst (blue), as well as for Epcam, Cd44, Cd24, and Aldh1a1 (green) of PanINs in KF mice (left panels) and PDAC in KPF mice (right panels). Scale bar, 50 μm.

The online version of this article includes the following source data for figure 1:

**Source data 1.** This spreadsheet contains the source data for *Figure 1D*.
**Source data 2.** This spreadsheet contains the source data for *Figure 1G*.
**Source data 3.** This spreadsheet contains the source data for *Figure 1H*.
**Source data 4.** This spreadsheet contains the source data for *Figure 1I*.
**Source data 5.** This spreadsheet contains the source data for *Figure 1J*.

tamoxifen administration, a few Dclk1[+] cells, which expressed EGFP encoded in *Dclk1*[CreERT2-IRES-EGFP] knock-in allele, were found in these PanINs (*Figure 2C–E*). Strikingly, 28 days after tamoxifen injection, we observed a significant number of PanIN lesions in which a majority of PanIN cells were labeled with EGFP (*Figure 2F*). The proportion of EGFP[+] cells, indicating a Dclk1[+] cells and their descendant cells, in PanIN cells increased from 5.43 ± 0.84% up to 35.4 ± 2.03% (*Figure 2G*, *Figure 2—source data 1*). Among EGFP[+] PanIN cells (the lineages of Dclk1[+] cells), there were a small subset of Dclk1-expressing cells 28 days after tamoxifen injection (*Figure 2H*). These data suggested that Dclk1[+] stem cells supply Dclk1[−] descendant cells in established PanINs and maintain these lesions.

## Dclk1[+] PDAC cells maintained mouse PDACs

We examined whether there is a potential leakiness of Cre expression in *Dclk1*[creERT2-IRES-EGFP] mice. We first examined whether the EGFP expression incorporated in the *Dclk1*[CreERT2-IRES-EGFP] allele coincides with the expression of Dclk1 before tamoxifen administration. Dclk1 staining was completely consistent with that of EGFP in normal pancreas and pancreatic epithelium irrespective of Kras and/or p53 mutation status (*Figure 3—figure supplement 1A*). Therefore, *Dclk1*[creERT2-IRES-EGFP] mice had no leakiness of Cre expression without tamoxifen administration. Using the *Rosa26*[mTmG] reporter allele, we next investigated whether CreER driven recombination occurred in Dclk1[−] cells of pancreatic epithelium in *Dclk1*[creERT2-IRES-EGFP] mice after tamoxifen administration. GFP[+] cells in Dclk1[−] cells were hardly seen (in less than 0.01% of Dclk1[−] cell population) in pancreatic epithelium of the mice with no gene alteration, with *Kras* single mutation or with both *Kras* mutation and *p53* deletion on day 1 and day 3 after tamoxifen administration (*Figure 3—figure supplement 1B*, *Figure 3—figure supplement 1—source data 1*). Therefore, we concluded that CreER driven recombination is specific to Dclk1[+] cells in *Dclk1*[creERT2-IRES-EGFP] mice after tamoxifen treatment.

Given that Dclk1[+] PanIN cells contributed to maintain PanINs, we next investigated by lineage tracing whether Dclk1[+] PDAC cells supply descendant PDAC cells also in PDACs. To this end, we employed a similar strategy and generated *Dclk1*[CreERT2-IRES-EGFP/+]; *Rosa26*[mTmG/+]; *Pdx1-Flp; Kras*[FSF-G12D/+]; *Trp53*[frt/frt] (DRKPF) mice (*Figure 3A*). In this model, the activation of flippase-*FRT* system induces oncogenic Kras[G12D] and Trp53 deletion in pancreatic progenitor cells, which results in the development of Tomato-labeled PDACs including a small number of Dclk1[+]/EGFP[+] cells (*Figure 3B*). Further Cre recombination by tamoxifen administration switches Tomato to EGFP in potential descendent cells of Dclk1[+] cells in PDACs, which allows to trace the lineages of Dclk1[+] cells in PDACs, as we did in PanINs (*Figure 3B*). Within 8 weeks of age, DRKPF mice developed PDAC, in which Tomato red protein was expressed by all tumor cells. Before tamoxifen injection, very few Dclk1[+]/EGFP[+] cells were observed in PDACs developed in DRKPF mice (*Figure 3C–E*). Two weeks after tamoxifen injection, Dclk1 lineage-positive (EGFP[+]) PDAC area dramatically increased (*Figure 3F*). We confirmed this result within Krt19[+] PDACs; very few EGFP[+] PDAC epithelial cells were observed within Krt19[+] tumor areas before tamoxifen injection (*Figure 3G*). The number of Dclk1 lineage-positive (EGFP[+]) cells increased within PDAC epithelium 14 days after tamoxifen

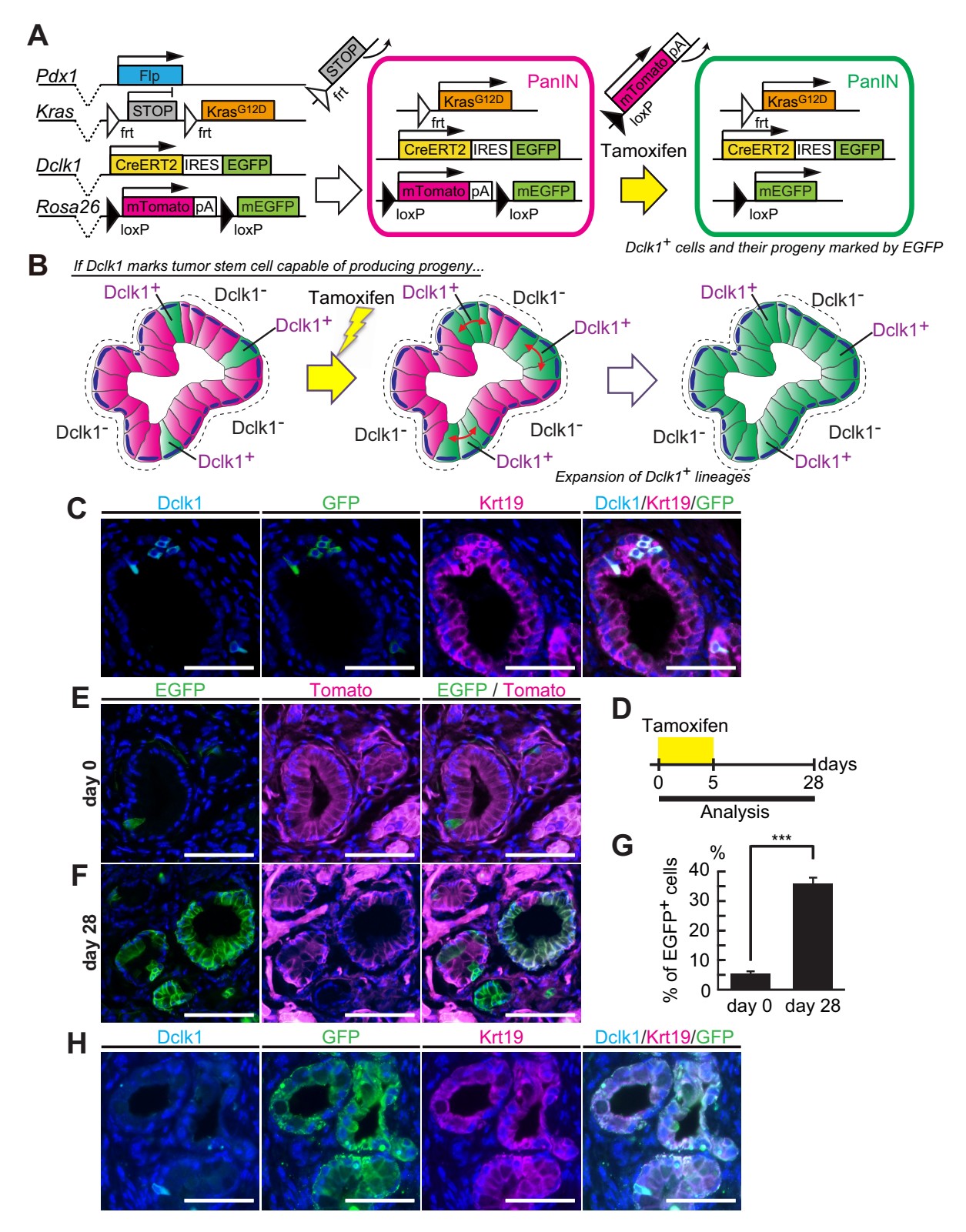

**Figure 2.** Dclk1+ PanIN cells supply descendant PanIN cells. (A) The scheme of *Dclk1^CreERT2-IRES-EGFP/+*; *Rosa26^mTmG/+*; *Pdx1-Flp*; *Kras^FSF-G12D/+* (DRKF) mouse constructs, flippase-mediated Kras activation, and CreERT2-driven reporter recombination. (B) The scheme of lineage tracing of Dclk1+ cells in established PanINs. Before tamoxifen administration, there were a small subset of Dclk1+/EGFP+ cells (left panel). After tamoxifen administration, if Dclk1+ cells are stem cells, EGFP+ progeny cells expand in the PanINs (middle and right panels). (C) Immunofluorescence staining for Dclk1 (cyan), GFP

*Figure 2 continued on next page*

*Figure 2 continued*

(green), Krt19 (magenta), and Hoechst (blue) of PanINs developed in DRKF mice before tamoxifen administration. Scale bars, 50 μm. (**D**) Experimental strategy of Cre-mediated lineage tracing in PanINs developed in 3-month-old DRKF mice. (**E and F**) Representative fluorescence microscopy images for EGFP encoded in *Dclk1^CreERT2-IRES-EGFP* knock-in allele (green), Tomato (magenta), and Hoechst (blue) in sections of PanINs developed in DRKF mice. (**E**) Before tamoxifen injection (day 0), EGFP (green) was expressed in Dclk1+ cells among Tomato-expressed PanIN cells (magenta). (**F**) After tamoxifen injection (day 28), the progeny of Dclk1+ cells expressed EGFP (green) and non-progeny cells still expressed Tomato (magenta). Scale bar, 50 μm. (**G**) Quantification of EGFP+ PanIN cells formed in DRKF mice before (day 0) and 28 days after tamoxifen injection (day 28). The number of PanIN lesions was 85 and 141 per mouse in day 0 and day 28 respectively (mean ± SEM; day 0, *n* = 6, left bar; day 28, *n* = 7, right bar; *n*: number of mice). Statistical significance of the difference is indicated as ***p<0.001, Student's *t*-test. (**H**) Immunofluorescence staining for Dclk1 (cyan), GFP (green), Krt19 (magenta), and Hoechst (blue) of PanINs developed in DRKF mice 28 days after tamoxifen administration. Scale bars, 50 μm.

The online version of this article includes the following source data for figure 2:

**Source data 1.** This spreadsheet contains the source data for *Figure 2G*.

injection (*Figure 3H*). EGFP+ area increased from 0.040 ± 0.005% on day 0 before tamoxifen injection to 54.6 ± 5.12% of Krt19+ PDAC area 14 days after tamoxifen injection (*Figure 3I*, *Figure 3— source data 1*). There were very few Dclk1+ cells within EGFP+ PDAC tumor cells 14 days after tamoxifen injection (*Figure 3J*). These results clearly demonstrated that a small subset of Dclk1+ PDAC cells continuously supply descendant PDAC cells and maintain also the PDACs that were already established in DRKPF mice.

## Dclk1+ cells supplied progeny in PDAC-derived tumor spheroids

Next, the time course of PDAC development was examined using a 3D culture method of tumor spheroids generated from PDACs of DRKPF mice. In those spheroids, Dclk1 was expressed only in a small fraction of tumor cells (*Figure 3—figure supplement 2A*). After 4-hydroxytamoxifen (4-OHT) treatment, the proportion of EGFP+ cells in PDAC-derived spheroids increased at 1, 2, and 3 days after the addition of 4-OHT (*Figure 3—figure supplement 2B and C*). The number of EGFP+ cells increased significantly from 1.05 ± 0.001% to 51.3 ± 0.05% 3 days after 4-OHT administration (*Figure 3—figure supplement 2D*) in which parental Dclk1+ cells were sparsely observed (*Figure 3—figure supplement 2E*). These data further supported our notion that Dclk1 is expressed in PDAC stem cells that continuously supply descendant PDAC cells also in the 3D-spheroid model.

## Dclk1+ cells supplied progeny in liver tumors from splenic transplantation

Next, we examined the stem cell potential of Dclk1+ PDAC cells in metastatic sites by applying an experimental model of metastatic pancreatic tumors. PDAC cells from DRKPF spheroids were injected into the spleen of BALB/c-nu mice. Eight weeks after splenic injection, liver tumors were found in 40% of BALB/c-nu mice (*Figure 4A*). Liver metastatic tumors preserved histological features of the primary PDACs (*Figure 4B and C*). Lineage tracing experiments were performed by administrating tamoxifen started at 8 weeks after splenic injection (*Figure 4D*). Before tamoxifen administration, a few Dclk1+/EGFP+ cells were found in these metastatic Krt19+ liver tumors (*Figure 4D–F*). At this point, only Dclk1+ cells were labeled by EGFP and most cells were tomato+ indicating Dclk1 lineage negative. After tamoxifen injection, EGFP+ Dclk1 lineage area started to expand and replaced tomato+ tumor areas, and the proportion of EGFP+ Dclk1 lineage area increased up to 32.1 ± 6.93% on day 14 from 0.014 ± 0.002% on day 0 (*Figure 4G and H*, *Figure 4—source data 1*). We observed that there were very few parental Dclk1+ cells within EGFP+ metastatic tumors, suggesting that most EGFP+ cells were descendant of Dclk1+ cells (*Figure 4I*). This striking data indicated that Dclk1+ PDAC cells show a functional stem cell activity even in metastatic liver tumors as well as in primary pancreatic lesions.

## Live imaging verified that Dclk1+ tumor cells supplied their progeny in the same PanIN and PDAC

To further strengthen lineage tracing data in vivo, we developed live imaging system to visualize the Dclk1+ cell lineage within the same PanIN and PDAC in the same individuals by introducing an abdominal imaging window (AIW; *Figure 5A*; *Alieva et al., 2014*; *Ritsma et al., 2013*) and two-

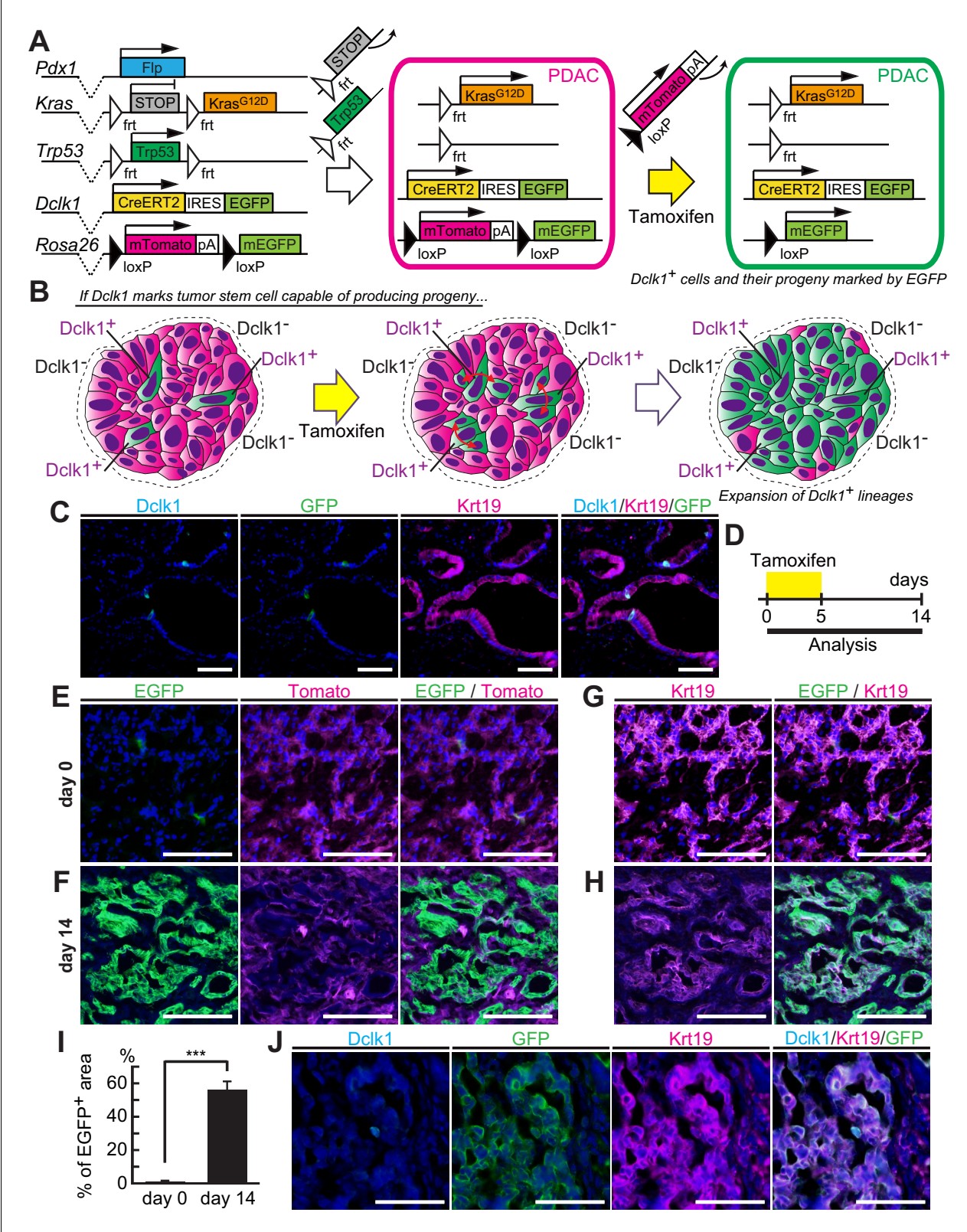

**Figure 3.** Dclk1+ pancreatic ductal adenocarcinoma (PDAC) cells supply descendant PDAC cells in vivo. (A) The scheme of *Dclk1*<sup>CreERT2-IRES-EGFP/+</sup>; *Rosa26*<sup>mTmG/+</sup>; *Pdx1-Flp; Kras*<sup>FSF-G12D/+</sup>; *Trp53*<sup>frt/frt</sup> (DRKPF) mouse constructs, flippase-mediated Kras activation and Trp53 deletion, and CreERT2-driven reporter recombination. (B) The scheme of lineage tracing of Dclk1+ cells in established PDAC. The flippase-*FRT* system produced Tomato+ PDACs including a small number of Dclk1+/EGFP+ cells (left panel). After tamoxifen administration, if Dclk1+ cells are PDAC stem cells, they supply

*Figure 3 continued on next page*

*Figure 3 continued*

EGFP$^+$ descendant PDAC cells in established PDACs (middle and right panels). (C) Immunofluorescence staining for Dclk1 (cyan), GFP (green), Krt19 (magenta), and Hoechst (blue) of PDACs developed in DRKPF mice before tamoxifen administration. Scale bars, 50 μm. (D) Experimental strategy of Cre-mediated lineage tracing in PDACs developed in 8-week-old DRKPF mice. (E and F) Representative fluorescent microscopy images for EGFP encoded in *Dclk1$^{CreERT2-IRES-EGFP}$* knock-in allele (green), Tomato (magenta), and Hoechst (blue) of sections of PDACs developed in DRKPF mice. (E) Before tamoxifen injection (day 0), EGFP encoded in *Dclk1$^{CreERT2-IRES-EGFP}$* knock-in allele was expressed in Dclk1$^+$ cells (green) among Tomato expressed PDAC cells (magenta). (F) After tamoxifen injection (day 14), the progeny of Dclk1$^+$ cells expressed EGFP (green) and non-progeny cells still expressed Tomato (magenta). Scale bar, 50 μm. (G and H) Overlay image of Krt19 staining and direct fluorescence observation of endogenous EGFP in same sections illustrated in E (G) and F (H). (I) Quantification of EGFP$^+$ area in PDACs developed in DRKPF mice before (day 0) and 14 days after tamoxifen injection (day 14). In DRKPF mice, a large tumor that replaced the entire pancreas was developed. One tumor was observed per mouse (mean ± SEM; day 0, *n* = 6, left bar; day 14, *n* = 6, right bar; *n*: number of mice). Statistical significance of the difference is indicated as ***p<0.001, Student's *t*-test. (J) Immunofluorescence staining for Dclk1 (cyan), GFP (green), Krt19 (magenta), and Hoechst (blue) of PDAC developed in DRKPF mice 14 days after tamoxifen administration. Scale bars, 50 μm.

The online version of this article includes the following source data and figure supplement(s) for figure 3:

**Source data 1.** This spreadsheet contains the source data for *Figure 3I*.
**Figure supplement 1.** *Dclk1$^{creERT2-IRES-EGFP}$* mice had no leakiness of Cre expression.
**Figure supplement 1—source data 1.** Analysis of the leak of Dclk1$^{CreERT2}$.
**Figure supplement 2.** Dclk1$^+$ cells supplied progeny in pancreatic ductal adenocarcinoma (PDAC) spheroids.
**Figure supplement 2—source data 1.** Lineage tracing of Dclk1$^+$ cells in established spheroids from pancreatic ductal adenocarcinomas (PDACs) of DRKPF mice.

photon excitation microscopy (TPEM). Mouse pancreas was pulled out of the peritoneal cavity and observed through AIW (*Figure 5B–D*).

For live mouse imaging, 6-week-old *Dclk1$^{CreERT2-IRES-EGFP/+}$*; *Rosa26$^{mTmG/+}$*; *Pdx1-Flp*; *Kras$^{FSF-G12D/+}$* (DRKF) mice received caerulein treatment in order to accelerate the development of PanINs (*Figure 5E*). Several EGFP$^+$ cells within Tomato$^+$ PanIN cells were observed through AIW, when an AIW was installed on the following day of the final tamoxifen injection (day 3, *Figure 5F*). At 10 days after AIW installation and the final tamoxifen injection, almost all PanIN cells expressed EGFP in the same PanIN lesions of the same live DRKF mouse (day 13, *Figure 5G*).

We also performed the time course observation of PDAC lesions developed in 6-month-old *Dclk1$^{CreERT2-IRES-EGFP/+}$*; *Rosa26$^{mTmG/+}$*; *Pdx1-Flp*; *Kras$^{FSF-G12D/+}$*; *Trp53$^{frt/+}$* (heterozygous DRKPF) mice. A PDAC nodule in DRKPF mice was observed through AIW (*Figure 5D*) by using TPEM. On the following day of the final tamoxifen injection, EGFP$^+$ PDAC cells were scarcely observed (day 3, *Figure 5H and I*). At 4 days after the final tamoxifen injection, numerous PDAC cells were EGFP$^+$ in the same live mouse (day 7, *Figure 5J*). Histologically, the vast majority of Krt19$^+$ cells of PDAC expressed EGFP (*Figure 5K*). These data clearly demonstrated that Dclk1$^+$ tumor cells supply descendant cells in both established PanIN and PDAC lesions in live mouse.

## Dclk1$^+$ PDAC cells possessed remarkable spheroid- and in vivo tumor-forming potentials

To evaluate the spheroid-forming potential of Dclk1$^+$ PDAC cells, Dclk1$^+$ and Dclk1$^-$ PDAC cells were collected by FACS from PDACs of DKPF mice (*Figure 6A*). PDAC cells were sorted by using an eFluor450-conjugated antibody against Epcam, because Epcam was expressed in the majority of PDAC cells (*Figure 1I and J*). Dclk1$^+$ cells were sorted as EGFP$^+$ cells within Epcam$^+$ cells (*Figure 6B*). Very few EGFP$^+$ cells (0.3% of Epcam$^+$ cells) and a large amount of EGFP$^-$ cells were collected (*Figure 6B*). EGFP$^+$ or EGFP$^-$ cells were suspended in growth-factor-reduced Matrigel at a density of 100 cells per well (*Figure 6A*). Whereas EGFP$^-$ PDAC cells formed almost no spheroids, EGFP$^+$ PDAC cells efficiently formed many large spheroids (*Figure 6C–F*), indicating that Dclk1$^+$ PDAC cells have a high spheroid-forming potential.

In addition, to further confirm the stemness of Dclk1$^+$ PDAC cells, Dclk1$^+$ and Dclk1$^-$ PDAC cells collected from DKPF mice by FACS were transplanted subcutaneously into the flank of NOD/SCID mice. Subcutaneous tumors were formed from either 100, 500, or 1000 Dclk1$^+$ PDAC cells in a sub-set of mice, whereas tumors were never formed even when 10,000 Dclk1$^-$ PDAC cells were transplanted (*Figure 6G and H*). The subcutaneous tumors displayed similar histological appearance to primary PDAC (*Figure 6I and J*), and a very small number of Dclk1$^+$ PDAC cells were observed in

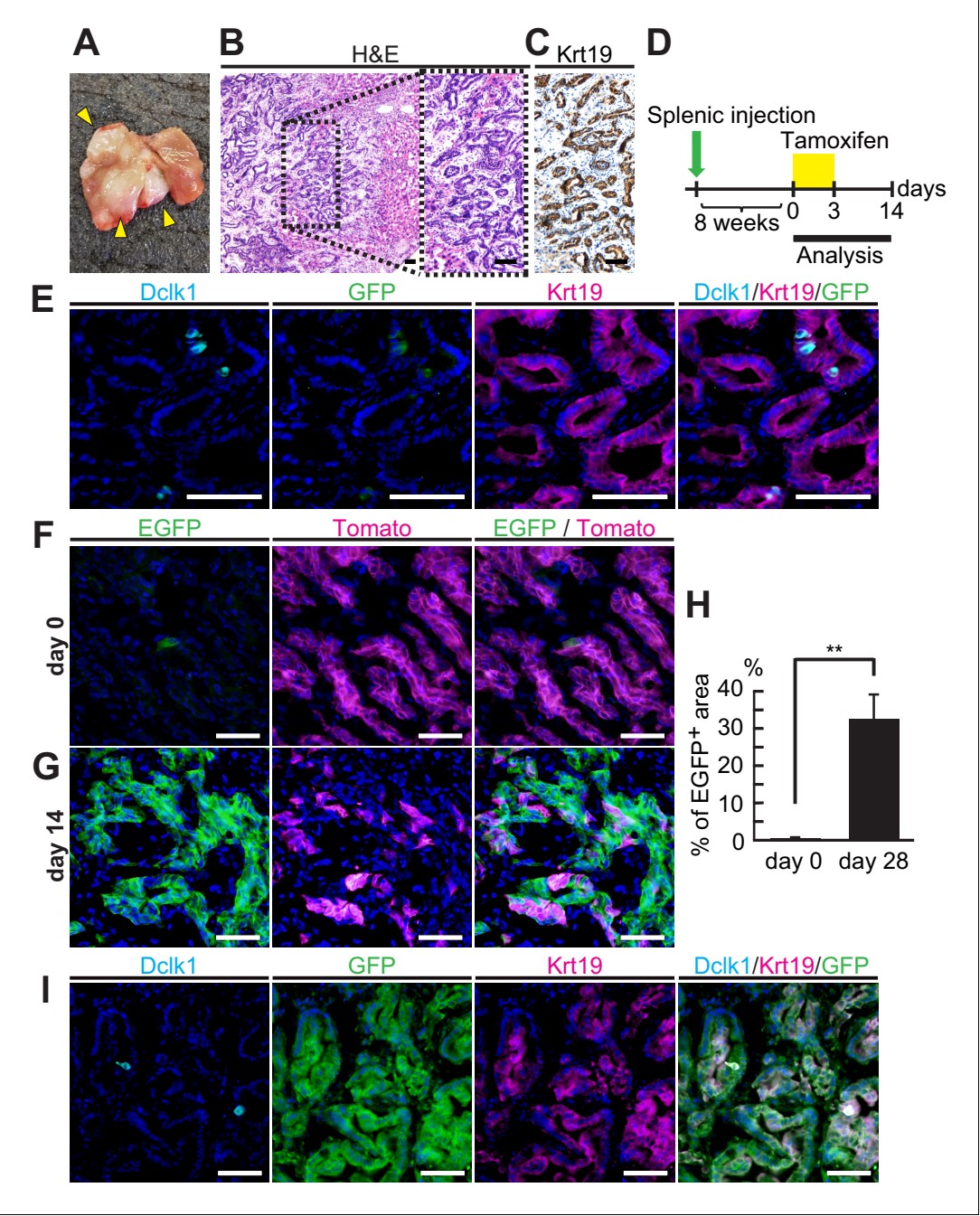

**Figure 4.** Dclk1+ cells supplied progeny in liver tumors from splenic transplantation. (**A**) Macroscopic image of liver tumors 2 months after splenic injection of pancreatic ductal adenocarcinoma (PDAC) spheroids derived from DRKPF mice. (**B and C**) Histological analysis of the metastatic liver tumors. (**B**) Hematoxylin and Eosin staining. (**C**) Immunostaining for Krt19. Scale bars, 50 μm. (**D**) Experimental strategy of Dclk1-Cre-mediated lineage tracing in established metastatic liver tumors. (**E**) Immunofluorescence staining for Dclk1 (cyan), GFP (green), Krt19 (magenta), and Hoechst (blue) of metastatic liver tumors developed by splenic injection of PDAC spheroids derived from DRKPF mice before tamoxifen administration. Scale bars, 50 μm. (**F and G**) Representative fluorescence microscopy images for EGFP encoded in *Dclk1^CreERT2-IRES-EGFP* knock-in allele (green), Tomato (magenta), and Hoechst (blue) in sections of metastatic liver tumors. (**F**) Before tamoxifen injection (day 0), EGFP encoded in *Dclk1^CreERT2-IRES-EGFP* knock-in allele was expressed in Dclk1+ cells (green) among Tomato-expressed tumor cells (magenta). (**G**) After tamoxifen injection (day 14), the progeny of Dclk1+ cells expressed EGFP (green) and non-progeny cells still expressed Tomato (magenta). Scale bar, 50 μm. (**H**) Quantification of EGFP+ area in liver tumor area before (day 0) and 14 days after tamoxifen injection (day 14, mean ± SEM; day 0, *n* = 6, left bar;

*Figure 4 continued on next page*

*Figure 4 continued*

day 14, *n* = 6, right bar; *n*: number of liver tumors, five mice in each groups). Statistical significance of the difference is indicated as **p<0.01, Student's *t*-test. (I) Immunofluorescence staining for Dclk1 (cyan), GFP (green), Krt19 (magenta), and Hoechst (blue) of metastatic liver tumors developed by splenic injection of PDAC spheroids derived from DRKPF mice. Scale bars, 50 μm.

The online version of this article includes the following source data for figure 4:

**Source data 1.** Lineage tracing of Dclk1$^+$ cells in established mouse metastatic liver tumors.

those tumors as well as in original PDACs (*Figure 6K*). These subcutaneous tumors increased in size over time (*Figure 6L*, *Figure 6—source data 1*). Furthermore, a thousand of Dclk1$^+$ and Dclk1$^-$ tumor cells collected from primary subcutaneous tumors were re-transplanted into the flank of other NOD/SCID mice. Re-transplanted Dclk1$^+$ cells formed subcutaneous tumors in 60% (3 out of 5) of mice, whereas Dclk1$^-$ cells developed no tumors (0 out of 5) (*Figure 6M*). Histologically, passaged tumors were also indistinguishable from primary subcutaneous tumors and original PDACs (*Figure 6N and O*), and a very small number of Dclk1$^+$ tumor cells were observed in those tumors (*Figure 6P*). These data further confirmed that Dclk1 marks PDAC stem cells with a tumor-forming ability in vivo.

## Dclk1$^+$ PDAC cells possessed EMT-, invasiveness-, and stemness-associated gene expression signature in mouse and human PDACs

To determine the molecular characteristics of Dclk1$^+$ PDAC cells, we performed microarray analysis on FACS-sorted Dclk1$^+$ and Dclk1$^-$ PDAC cells obtained from *Dclk1$^{CreERT2-IRES-EGFP/+}$*; *Pdx1-Flp*; *Kras$^{FSF-G12D/+}$*; *Trp53$^{frt/frt}$* (DKPF) mice (GSE139167), which revealed 4395 differentially expressed genes (p<0.01; 2171 genes upregulated and 2224 genes downregulated in Dclk1$^+$ PDAC cells; *Figure 7A*). Dclk1$^+$ PDAC cells showed significantly higher expression levels of epithelial mesenchymal transition (EMT)-associated genes such as *Vim*, *Snai1*, *Snai2*, *Twist1*, and *Twist2* (*Mani et al., 2008*; *Yang et al., 2004*; *Ansieau et al., 2008*), and pancreatic CSC marker *Aldh1a1* (*Kim et al., 2011*; *Rasheed et al., 2010*), which is consistent with the intimate relationship of CSC characteristics with EMT phenotype (*Figure 7A*).

Gene ontology (GO) enrichment analysis and pathway analysis were performed on The Database for Annotation, Visualization and Integrated Discovery (DAVID) using these 2171 genes highly expressed in Dclk1$^+$ PDAC cells. In the GO enrichment analysis using GO_Biological Process data set, 454 GO terms were enriched in Dclk1$^+$ PDAC cells and GO terms associated with angiogenesis and EMT were included in top 100 GO terms (*Supplementary file 1*). In KEGG data set, Dclk1$^+$ PDAC cells were significantly enriched with 58 pathways, which included those associated with stemness and drug resistance (*Supplementary file 2*). Gene set enrichment analysis (GSEA) identified 'Multicancer invasiveness signature', 'Stem cell up', and 'Epithelial mesenchymal transition' as significantly upregulated signatures in Dclk1$^+$ PDAC cells (*Figure 7B*). These results further support our notion that mouse Dclk1$^+$ PDAC cells possess characteristics of CSC and CSC-related signatures such as EMT, invasiveness, and drug resistance as described in the previous reports (*Mani et al., 2008*; *Meirelles et al., 2012*; *Li et al., 2008*).

To examine the role of DCLK1$^+$ cells in human PDACs, we investigated human PDAC data sets (GSE16515; 36 samples, GSE32676; 25 samples, GSE36924; 91 samples, and TCGA_PAAD; 183 samples). GSEA revealed the positive enrichment of EMT, invasiveness, and stem cell signatures in *DCLK1*-high PDACs when compared to *DCLK1*-low PDACs (*Figure 7C*). In addition, we observed the strong correlation of *VIM*, the EMT-associated gene, with *DCLK1* expression and inverse correlation of *CDH1*, associated with epithelial state (*Figure 7D*).

GO analysis and pathway analysis were performed using DAVID on genes highly expressed (p<0.05) in the *DCLK1* high expression group of these human PDAC data sets. As in the mouse data, GO terms related to angiogenesis ('angiogenesis' GSE16515, GSE32676, TCGA_PAAD) and pathway related to stemness ('Signaling pathways regulating pluripotency of stem cells' GSE36924) were enriched in the *DCLK1* high expression group (*Supplementary file 3*).

These data suggest that DCLK1$^+$ PDAC cells harbor the characteristics of potential CSCs also in human PDAC as well as in mouse PDAC.

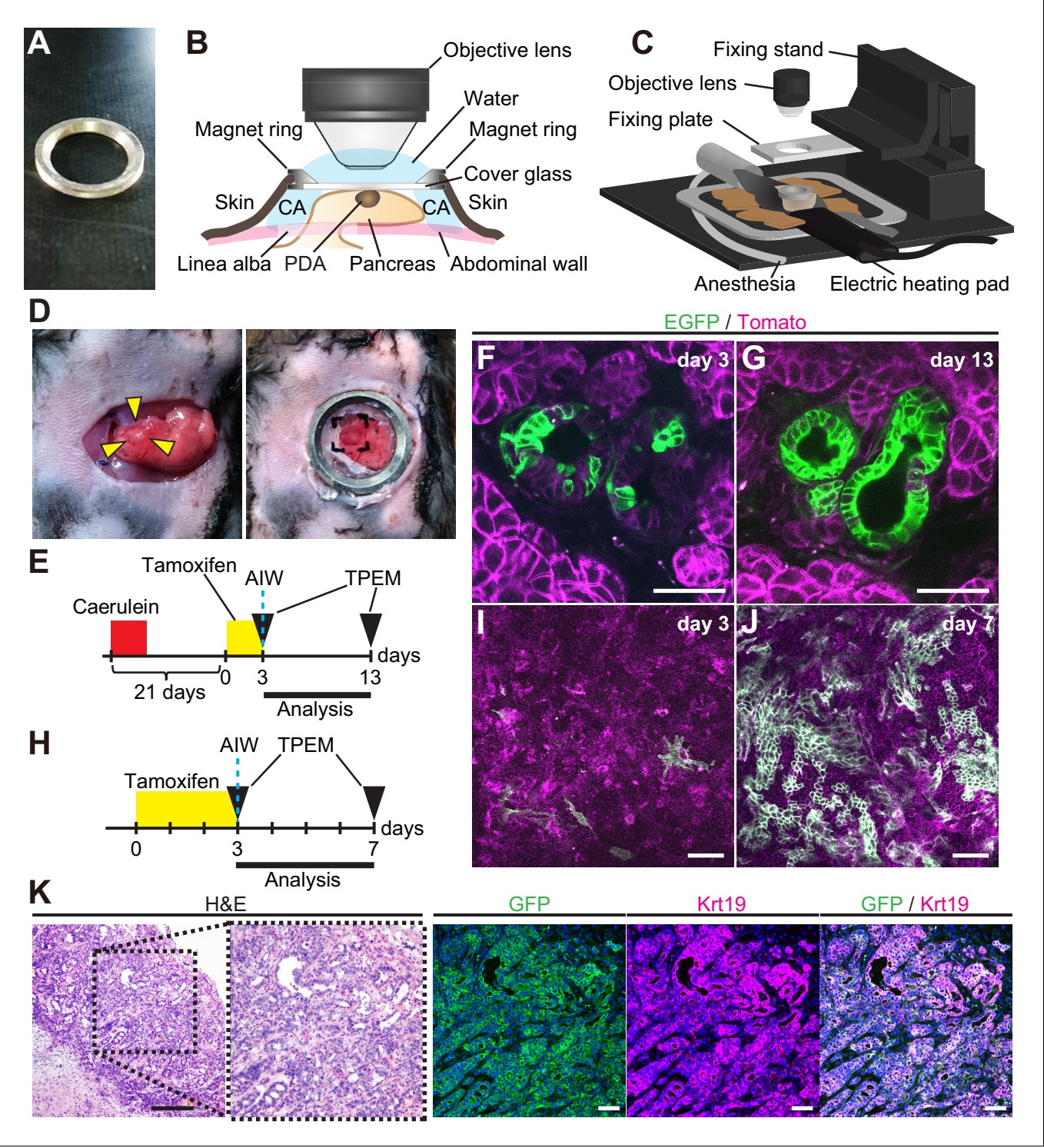

**Figure 5.** Longitudinal imaging of PanINs and pancreatic ductal adenocarcinomas (PDACs) in live mice indicate Dclk1+ PDAC cells supply progeny PDAC cells. (A) Photo of an abdominal imaging window (AIW) composed of a custom-made magnet ring and 12 mm cover glass. (B) Scheme of the microscopic observation of the pancreas through an AIW. CA: cyanoacrylate. (C) Layout of the system for mouse live pancreas imaging. (D) Macroscopic images of pancreas with a nodule (arrowhead) transferred on the peritoneum (left) and observed through AIW (right). (E) Protocol of live imaging of PanINs formed in DRKF mice. The mice were treated with caerulein to accelerate PanIN formation. (F and G) Live imaging of PanINs formed in DRKF mice. (F) On the day of AIW installation, the following day of the final tamoxifen injection (day 3), several EGFP+ cells were observed through AIW (green) among Tomato-expressed tumor cells (magenta). (G) Ten days after AIW installation (day 13), almost all PanIN cells expressed EGFP in the same PanIN lesions of the same live DRKF mouse. Scale bar, 50 μm. (H) Protocol of live imaging of PDACs formed in DRKPF mice. (I and J) Live

*Figure 5 continued on next page*

*Figure 5 continued*

imaging of PDACs formed in DRKPF mice. (I) On the day of AIW installation, the following day of the final tamoxifen injection (day 0), EGFP⁺ PDAC cells were scarcely observed (green). (J) Seven days after tamoxifen injection, numerous PDAC cells were EGFP⁺ in the same live mouse (green). Scale bars, 50 μm. (K) Representative images of H and E staining and immunofluorescence staining for GFP, Krt19, and Hoechst of PDAC after live imaging. Scale bar, 50 μm.

## Discussion

CSCs have been identified and intensively studied in many kinds of cancers (*Visvader and Lindeman, 2008*; *Rosen and Jordan, 2009*). In regard to PDAC, several markers of CSCs have been identified by transplantation and spheroid forming assays (*Bailey et al., 2014*; *Westphalen et al., 2016*; *Lee et al., 2008*; *Li et al., 2007*; *Kim et al., 2011*; *Rasheed et al., 2010*). However, transplantation assays are performed in the context of non-natural environments that contain different niche factors from those in original tumor sites; the use of immunocompromised mice for transplantation of sorted cell populations adds more complexity and artificiality in the model. Regarding this, lineage tracing is one of the most rigorous and robust tools to elucidate the natural activity and behavior of CSCs within their native environment in vivo, because it can directly visualize the hierarchical structures without affecting microenvironmental factors such as stem cell niche and surrounding immune/stromal cells (*Kretzschmar and Watt, 2012*). However, so far, no longitudinal lineage tracing of pancreatic tumor/CSCs within established tumors has been performed in vivo. Furthermore, live imaging techniques to observe the expansion of tumor cells within the same tumors in the same mice have not yet been developed, because they have been technically challenging. In this study, we established a novel system to trace the dynamics of CSCs by utilizing an inducible dual-recombinase system that combined flippase-*FRT* and Cre-*loxP* recombinations in mice. Furthermore, we here performed for the first time the genetic lineage tracing with live imaging within primary PDAC and its precursor lesions as well as metastatic lesions that directly provided crucial stem cell activity of Dclk1⁺ tumor cells in vivo.

A previous study showed that Dclk1⁺ cells can be PDAC initiating cells in the context of pancreatitis and that Dclk1⁺ cells have stem cell properties to sustain spheroid growth in vitro (*Westphalen et al., 2016*). Another study previously showed that DCLK1⁺ human PDAC cells have a high spheroid-forming potential in vitro (*Bailey et al., 2014*). Although these studies demonstrated a role of Dclk1⁺ cells in PDAC initiation in vivo and tumor stem cell properties of Dclk1⁺ cells in vitro, natural behavior and tumor/CSC activity of Dclk1⁺ cells within established tumors in vivo have been unclear. Our study anew established the functional role of Dclk1⁺ tumor cells for the maintenance and expansion of primary and metastatic PDAC and its precursor lesions in vivo.

The availability of mouse inducible dual-recombinase strains including flippase-*FRT* and Cre-*loxP* recombinations and the advancements in fluorescent microscopy allowed us to develop a powerful lineage tracing system. By using this system, we here demonstrated the three novel and quite important findings. First, we revealed that Dclk1⁺ PanIN and PDAC cells supplied descendant cells in established PanIN and PDAC within native environments in vivo, demonstrating the stem cell activity of Dclk1⁺ tumor cells within primary pancreatic malignancy. Second, we utilized live fluorescence imaging analysis that enabled longitudinal lineage tracing observations in the same live animals. Live two-photon excitation fluorescence imaging through an AIW enabled us to make deep-focus observations of live tissue. We have taken advantage of this powerful methodology to perform direct longitudinal lineage tracing of pancreatic CSCs in live mice. Our study successfully described the time course of PanIN/PDAC maintenance and expansion that critically proved the Dclk1⁺ cells as CSCs within established PDAC and its precursor lesions in live mice. Third, we also found that Dclk1⁺ PDAC cells supplied descendent PDAC cells even in metastatic liver tumors formed by splenic transplantation. Thus, our lineage tracing has provided the quite robust data to identify the stem cell activity of Dclk1⁺ cells within pancreatic neoplasia in all the progressive stages in vivo.

In this study, we observed EMT associated gene expression signature enriched in Dclk1⁺ tumor cells in both mouse and human PDACs. This is consistent with previous studies showing that signaling pathways that are involved in metastasis are upregulated in pancreatic CSCs (*Mani et al., 2008*). We also found that Dclk1⁺ PDAC cells have higher expression of Aldh1a1 compared with Dclk1⁻

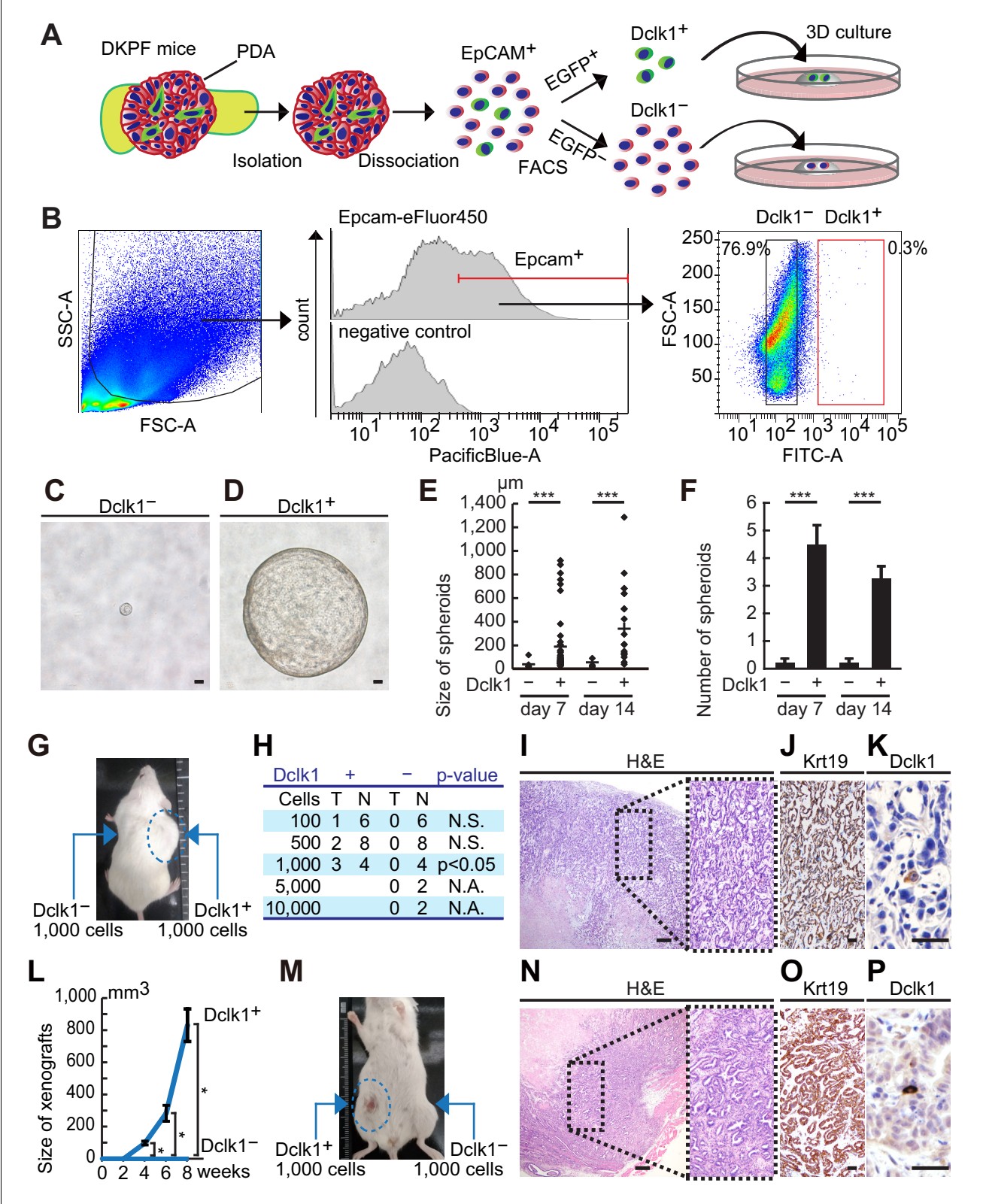

**Figure 6.** Dclk1[+]pancreatic ductal adenocarcinoma (PDAC) cells have remarkable spheroid- and tumor-forming potential. (A) Scheme of FACS and 3D culture of Dclk1[+] and Dclk1[−] PDAC cells from DKPF mice. (B) FACS-sorting of PDAC cells on the basis of Epcam and Dclk1 expression in DKPF mice. (C and D) Representative images of tumor spheroids derived from sorted Dclk1[−] (C) and Dclk1[+] (D) PDAC cells at day 7. (E) Size of spheroids were compared (mean ± SEM; Dclk1[+], $n$ = 5; Dclk1[−], $n$ = 5; $n$: number of mice). Statistical significance of the differences is indicated as ***$p<0.001$, Student's

*Figure 6 continued on next page*

*Figure 6 continued*

*t*-test. (**F**) The number of spheroids larger than 20 µm were compared (mean ± SEM; Dclk1⁺, *n* = 5; Dclk1⁻, *n* = 5; *n*: number of mice). Statistical significance of the differences is indicated as \*\*\*p<0.001, Student's *t*-test. (**G and H**) Tumor-forming assay of Dclk1⁺ or Dclk1⁻ PDAC cells. (**G**) Macroscopic image of the NOD/SCID mice after subcutaneous transplantation of Dclk1⁺ or Dclk1⁻ PDAC cells into the flank. A thousand of Dclk1⁺ PDAC cells developed subcutaneous tumors whereas Dclk1⁻ PDAC cells did not. (**H**) Dilution series showed 100, 500, or 1000 Dclk1⁺ PDAC cells developed subcutaneous tumors in 16.7%, 25.0%, and 75.0% of NOD/SCID mice, whereas same or larger numbers of Dclk1⁻ PDAC cells did not. Statistical significance of the differences is indicated as p<0.05, chi-squared test. (**I–K**) Histological analysis of primary xenografts derived from FACS-sorted Dclk1⁺ PDAC cells. (**I**) Hematoxylin and Eosin (H and E) staining. (**J**) Immunostaining for Krt19. (**K**) Immunostaining for Dclk1. Scale bar, 200 µm (**I**), 50 µm (**J and K**). (**L**) Increasing curve of subcutaneous tumor (mean ± SEM; Dclk1⁺, *n* = 3; Dclk1⁻, *n* = 4; *n*: number of mice). Statistical significance of the differences is indicated as \*p<0.05, Student's *t*-test. (**M**) Passaged tumor-forming assay of Dclk1⁺ or Dclk1⁻ xenograft cells collected by FACS. A thousand of Dclk1⁺ xenograft cells developed subcutaneous tumors whereas Dclk1⁻ xenograft cells did not. (**N–P**) Histological analysis of passaged xenografts derived from FACS-sorted Dclk1⁺ primary xenograft cells. (**N**) Hematoxylin and Eosin (H and E) staining. (**O**) Immunostaining for Krt19. (**P**) Immunostaining for Dclk1. Scale bar, 200 µm (**N**), 50 µm (**O and P**).

The online version of this article includes the following source data for figure 6:

**Source data 1.** Growth of pancreatic ductal adenocarcinoma (PDAC) xenograft.

cells. These results strongly suggest that Dclk1⁺ cells possessed CSC-like potentials not only in mouse PDAC but also in human PDAC.

Considering the clinical relevance, while this study has provided strong clues to understand how pancreatic neoplasia is maintained and expanded, there remains an important question whether specific ablation of Dclk1⁺ cells in established PanIN and PDAC lesions results in their regression in vivo. Future studies would be required to uncover the essential role of Dclk1⁺ tumor/CSCs by specific lineage ablation and to determine whether targeting Dclk1⁺ tumor cells could be a novel therapeutic approach against PDAC. In this respect, recent studies showed that selective ablation of LGR5⁺ colorectal CSCs led to tumor regression, which was followed by tumor regrowth upon treatment cessation (*de Sousa e Melo et al., 2017*; *Shimokawa et al., 2017*). Therefore, given that combined chemotherapy potentially targeted CSCs in the recent study (*Shimokawa et al., 2017*), our present study suggests that targeting Dclk1⁺ cells can be a novel therapeutic option for PDAC, although it might have to be combined with standard chemotherapy for the therapy to be effective for a long period.

In conclusion, by utilizing the novel system for genetic lineage tracing with live imaging, including the use of a dual-recombinase system that combined both flippase-*FRT* and Cre-*loxP* recombinations, we for the first time provided direct evidence for the crucial stem cell activity of Dclk1⁺ tumor cells within primary PDAC and its precursor lesions as well as metastatic lesions in vivo. These findings provide insights into understanding the biology of PDAC and novel therapeutic approaches against this dismal disease.

# Materials and methods

### Key resources table

| Reagent type (species) or resource | Designation | Source or reference | Identifiers | Additional information |
|---|---|---|---|---|
| Gene *Mus musculus* | *Dclk1* | NCBI Gene Database | NCBI Gene: 13175 | |
| Gene (*Homo sapiens*) | *DCLK1* | NCBI Gene Database | NCBI Gene: 9201 | |
| Genetic reagent (*Mus musculus*) | *Dclk1^CreERT2-IRES-EGFP* | Generated in our laboratory *Nakanishi et al., 2013* | N/A | |
| Genetic reagent (*Mus musculus*) | *Pdx1-Flp* | Saur D. *Schönhuber et al., 2014* | N/A | |
| Genetic reagent (*Mus musculus*) | *Kras^FSF-G12D* | Saur D. *Schönhuber et al., 2014* | N/A | |

*Continued on next page*

*Continued*

| Reagent type (species) or resource | Designation | Source or reference | Identifiers | Additional information |
|---|---|---|---|---|
| Genetic reagent (*Mus musculus*) | *Rosa26*$^{mTmG}$ | Saur D. *Schönhuber et al., 2014* | N/A | |
| Genetic reagent (*Mus musculus*) | *Trp53*$^{frt}$ | Jackson Laboratory *Schönhuber et al., 2014* | RRID:IMSR_JAX:017767 | |
| Genetic reagent (*Mus musculus*) | CAnN.Cg-Foxn1$^{nu}$/Crl | Charles river | N/A | BALB/c-nu |
| Genetic reagent (*Mus musculus*) | NOD.CB17-Prkdc$^{scid}$/J | Jackson Laboratory | RRID:IMSR_JAX:001303 | NOD scid |
| Antibody | Rabbit polyclonal anti-Dcamkl1 | Abcam | Cat#: ab31704 RRID:AB_873537 | 1:200 |
| Antibody | Goat polyclonal anti-Dcamkl1 | Santa Cruz Biotechnology | Cat#: sc46312 RRID:AB_2090091 | 1:50 |
| Antibody | Rat monoclonal anti-Cd24 | Abcam | Cat#: ab64064 RRID:AB_2291132 | 1:100 |
| Antibody | Rat monoclonal anti-Cd44 | Abcam | Cat#: ab119863 RRID:AB_10898986 | 1:100 |
| Antibody | Rat monoclonal anti-Cd326 | Thermo Fisher Scientific | Cat#: 13-5791-82 RRID:AB_1659713 | 1:100 |
| Antibody | Rabbit polyclonal anti-Aldh1a1 | Abcam | Cat#: ab23375 RRID:AB_2224009 | 1:100 |
| Antibody | Rat monoclonal anti-Cd326 (EpCAM) eFluor450 | Thermo Fisher Scientific | Cat#: 48-5791-82 RRID:AB_10717090 | 1:50 |
| Antibody | Goat polyclonal anti-GFP | Abcam | Cat#: ab6673 RRID:AB_305643 | 1:100 |
| Antibody | Mouse monoclonal anti-Cytokeratin | DAKO | Cat#: IR05361-2J RRID:AB_2868599 | 1:1 |
| Sequence-based reagent | Dclk1-CreERT2 Forward1 | This paper | PCR primers | CGAGCTGGACG GCGACGTAAACG |
| Sequence-based reagent | Dclk1-CreERT2 Forward2 | This paper | PCR primers | GATGGACTCAAG AAGATCTCC |
| Sequence-based reagent | Dclk1-CreERT2 Reverse | This paper | PCR primers | AGTGACCCTTAG TGACCCTTAGT |
| Sequence-based reagent | Pdx1-Flp Forward | Saur D. *Schönhuber et al., 2014* | PCR primers | AGAGAGAAAATTG AAACAAGTGCAGGT |
| Sequence-based reagent | Pdx1-Flp Reverse | Saur D. *Schönhuber et al., 2014* | PCR primers | CGTTGTAAGGG ATGATGGTGAACT |
| Sequence-based reagent | Kras Common Forward | Saur D. *Schönhuber et al., 2014* | PCR primers | CACCAGCTTCG GCTTCCTATT |
| Sequence-based reagent | Kras WT Reverse | Saur D. *Schönhuber et al., 2014* | PCR primers | AGCTAATGGCTC TCAAAGGAATGTA |
| Sequence-based reagent | Kras FSF MUT reverse | Saur D. *Schönhuber et al., 2014* | PCR primers | GCGAAGAGTTTG TCCTCAACC |

*Continued on next page*

*Continued*

| Reagent type (species) or resource | Designation | Source or reference | Identifiers | Additional information |
|---|---|---|---|---|
| Sequence-based reagent | p53-frt1 | Jackson Laboratory | PCR primers | CAAGAGAAC TGTGCCTAAGAG |
| Sequence-based reagent | p53-frt2 | Jackson Laboratory | PCR primers | CTTTCTAACAGC AAAGGCAAGC |
| Software, algorithm | Image J | National Institutes of Health | RRID:SCR_003070 | https://imagej.net/ |
| Software, algorithm | InSight DeepSee Laser | Spectra Physics | RRID:SCR_012362 | http://www.scienceexchange.com/facilities/multiphoton-microscopy-core-rochester |
| Software, algorithm | FlowJo | FlowJo, LLC | RRID:SCR_008520 | https://www.flowjo.com/ |

## Animal experiments

All the animal experiments were approved by the animal research committee of the Kyoto University and performed in accordance with Japanese government regulations and all the animals were maintained in a specific pathogen-free facility. All surgery was performed under Isoflurane anesthesia, and every effort was made to minimize suffering. The following mouse lines were used: $Dclk1^{CreERT2-IRES-EGFP}$ (*Nakanishi et al., 2013*), $Trp53^{frt}$ (Jackson Laboratory, Bar Harbor, ME), *Pdx1-Flp* (*Schönhuber et al., 2014*), $Kras^{FSF-G12D}$ (*Schönhuber et al., 2014*), $Rosa26^{mTmG}$ (*Schönhuber et al., 2014*), BALB/c-nu mice (CAnN.Cg-Foxn1nu/Crl; Charles river, Wilmington, MA), and NOD/SCID mice (NOD.CB17-Prkdcscid/J; Jackson Laboratory). Mice were crossed in a mixed background and no selection for a specific gender was done in this study. Tamoxifen (Sigma-Aldrich, St. Louis, MO) was dissolved in corn oil (Wako, Osaka, Japan) and administered intraperitoneally at a concentration of 2 mg/20 g body weight or subcutaneously at a concentration of 4 mg/20 g per injection. Acute pancreatitis was induced by injecting caerulein (Sigma-Aldrich) as described in a previous report (*Jensen et al., 2005*).

## Human PDACs specimens

Seven surgically resected specimens of pancreatic cancer tissues were obtained from patients who had been admitted to Kyoto University Hospital. Written informed consent was obtained from all patients and study protocol (#G1200-1) was approved by Ethics Committee of Kyoto University Hospital.

## Lineage tracing experiments

For experiments to investigate potential leakiness of Cre expression, 6- to 10-week-old $Dclk1^{CreERT2-IRES-EGFP/+}$; *Pdx1-Flp* (DF), $Dclk1^{CreERT2-IRES-EGFP/+}$; *Pdx1-Flp*; $Kras^{FSF-G12D/+}$ (DKF) and $Dclk1^{CreERT2-IRES-EGFP/+}$; *Pdx1-Flp*; $Kras^{FSF-G12D/+}$; $Trp53^{frt/frt}$ (DKPF) mice were used without tamoxifen administration. For experiments with tamoxifen administration, 4- to 5-week-old $Dclk1^{CreERT2-IRES-EGFP/+}$; $Rosa26^{mTmG/+}$; *Pdx1-Flp* (DRF), $Dclk1^{CreERT2-IRES-EGFP/+}$; $Rosa26^{mTmG/+}$; *Pdx1-Flp*; $Kras^{FSF-G12D/+}$ (DRKF) and $Dclk1^{CreERT2-IRES-EGFP/+}$; $Rosa26^{mTmG/+}$; *Pdx1-Flp*; $Kras^{FSF-G12D/+}$; $Trp53^{frt/frt}$ (DRKPF) mice, with few tumors developed, were used. These mice received a single intraperitoneal tamoxifen administration (2mg/20 g body weight) and were analyzed 1 and 3 days after tamoxifen treatment.

For genetic lineage tracing experiments, Cre-recombination in mice was activated with tamoxifen. For lineage tracing, mice were injected with five or three doses of 2 mg tamoxifen. To perform lineage tracing of spheroids, 4-hydroxytamoxifen (final concentration: $2.0 \times 10^{-6}$ M, 4-OHT, Sigma-Aldrich) and Hoechst (final concentration: 5.0 μg/ml, Thermo Fisher Scientific, Waltham, MA) were added to the medium in which PDAC spheroids derived from DRKPF ($Rosa26^{mTmG/+}$ or $Rosa26^{mTmG/mTmG}$) mice 5 days after the suspension and removed after 24 hr of exposure. For live imaging, mice

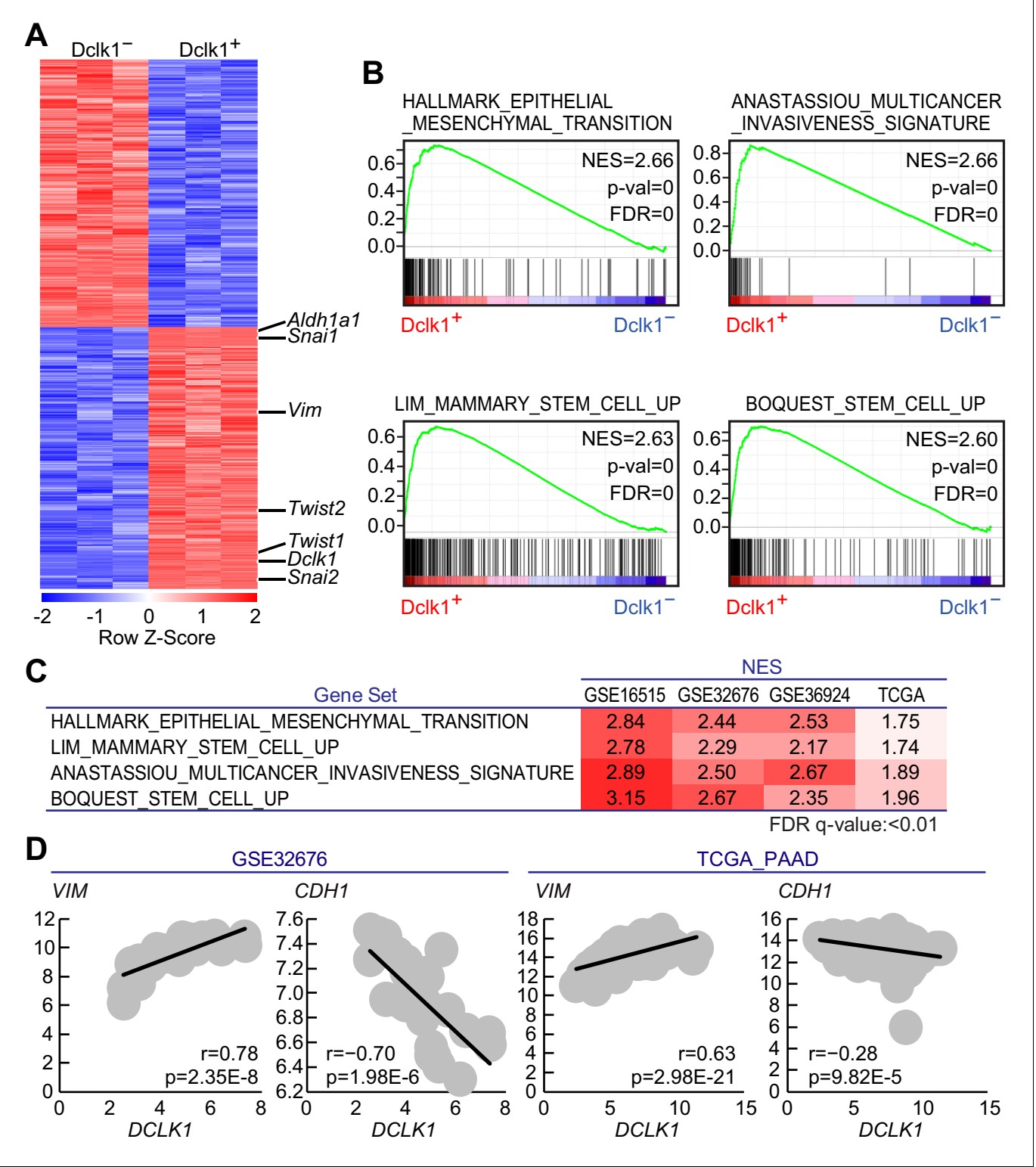

**Figure 7.** Gene expression profile revealed Dclk1+pancreatic ductal adenocarcinoma (PDAC) cells possessed cancer stem cell potential. (**A**) The heatmap of representative differentially expressed genes with a significant difference, p<0.01 (Student's *t*-test) in gene expression. (**B**) Gene set enrichment analysis (GSEA) in Dclk1+ versus Dclk1− PDAC cells. (**C**) Gene set enrichment analysis (GSEA) in *DCLK1*-high versus *DCLK1*-low PDACs in human PDAC data sets. (**D**) Correlation analysis with *VIM* or *CDH1* and *DCLK1* in human PDAC data sets. (r, Pearson correlation coefficient). Statistical significance of the differences is indicated as p<0.001, Student's *t* distribution.

were administered 4 mg of tamoxifen subcutaneously for three consecutive days – just before AIW installation and first observation by a TPEM.

## Immunohistochemistry

For immunohistochemistry, tissues were perfused and fixed in 4% paraformaldehyde/PBS, dehydrated into 70% ethanol, embedded in paraffin, and sectioned at 5 μm thickness. Paraffin-embedded sections were stained with hematoxylin and eosin. Antigen retrieval was performed by incubating sections in citric acid buffer (pH 6.0) or EDTA buffer (pH 8.0) for 15 min at 98°C. Blocking was performed by incubating sections in 2% BSA/PBS solution (Wako). The primary antibodies used in this study were as follows: rabbit anti-Dclk1 (1:200; Abcam, Cambridge, UK), goat anti-Dclk1 (1:50; Santa Cruz Biotechnology, Santa Cruz, CA), rat anti-Cd24 (1:100; Abcam), rat anti-Cd44 (1:100; Abcam), rat anti-Cd326 (1:100; Thermo Fisher Scientific), rabbit anti-Aldh1a1 (1:100; Abcam), goat anti-GFP (1:100; Abcam), and Envision FLEX-Cytokeratin (1:1; DAKO). Primary antibodies were incubated for 2 hr at room temperature or overnight at 4°C. Secondary antibodies were incubated for 1 hr at room temperature. For immunohistochemistry, slides were developed using EnVision kit (Dako, Glostrup, Denmark) followed by counterstaining with hematoxylin. For immunofluorescence staining, sections were nuclear stained with Hoechst (Thermo Fisher Scientific).

For direct microscopic observation of sections, tissues were perfused and fixed in 4% paraformaldehyde and 30% sucrose, frozen in FSC 22 Frozen Section Media and sectioned at 10 μm thickness. For quantification analysis, the numbers of EGFP$^+$ or Tomato$^+$ cells in PanINs were counted.

For direct microscopic observation combined with immunofluorescence, after quenching internal fluorescence by heat treatment, sections were incubated for 2 hr at room temperature with primary antibodies against Krt19 and for 1 hr at room temperature with secondary antibodies. For quantification analysis, EGFP$^+$ or Krt19$^+$ areas of PDAC were quantified using ImageJ software (National Institutes of Health, Bethesda, MD).

## Cell culture

PDAC tissue was freshly isolated, minced, suspended in 2.5 ml of the digestion buffer, and dissociated with a gentleMACS Dissociator (Miltenyi Biotec, Bergisch Gladbach, Germany) at 'm-Imp Tumor 02' setting. The material was incubated at 37°C for 15 min, further dissociated with gentleMACS Dissociator at 'm-Imp Tumor 03' setting, passed through 100 μm and 40 μm cell strainer, embedded in growth-factor-reduced Matrigel (BD Biosciences), and cultured in the culture medium. Digestion buffer consisted of HBSS without calcium, magnesium, and phenol red (Thermo Fisher Scientific) supplemented with 2.5 mg/ml collagenase D (Roche, Basel, Switzerland), 1.14 mg/ml Dispase (Thermo Fisher Scientific), 1 mg/ml DNaseI (Qiagen, Hilden, Germany), and 4% fetal bovine serum (Thermo Fisher Scientific). Culture medium consisted of Advanced DMEM/F-12 medium (Thermo Fisher Scientific) supplemented with ×100 GlutaMAX (Thermo Fisher Scientific) and 10% fetal bovine serum (Thermo Fisher Scientific) and 100 U/ml Penicillin–Streptomycin (Thermo Fisher Scientific). Cells were maintained at 37°C in 5% $CO_2$.

## Splenic injection of PDAC cells

For the lineage tracing analysis of liver metastasis, $10^5$ PDAC cells were injected from the lower pole of the spleen with a 25-gauge needle (*Kozlowski et al., 1984*).

## Abdominal imaging window

AIW was created from a custom-made magnet ring with a 12 mm round cover glass (Matsunami, Osaka, Japan) glued with cyanoacrylate (Konishi). AIW was glued to the pancreas and abdominal wall with jelly-like cyanoacrylate (Konishi). In order to observe the same place over time, cover glass was marked. Mice with inserted AIW were placed on an electric heating pad on a custom-designed fixing stand (*Sano et al., 2016*), pressed by a fixing plate attached with the fixing stand to keep the AIW level (*Takaoka et al., 2016*), and the pancreas was observed through the AIW by TPEM.

## Live imaging of living mice

We used an FV1200MPE-BX61WI upright microscope (Olympus, Tokyo, Japan) equipped with a 25×/1.05 water-immersion objective lens (XLPLN25XW-MP; Olympus) and an InSight DeepSee

Ultrafast Laser (Spectra Physics, Mountain View, CA). The excitation wavelength for EGFP and Tomato was 840 nm. We used an IR-cut filter, BA685RIF-3, two dichroic mirrors, DM505 and DM570 (Semrock, Rochester, NY), and two emission filters, BA520-560 (Olympus) — for EGFP and FF01-647/57 (Semrock) — for Tomato, respectively. Laser power was set to 10–14% for the observation of the pancreas. In order to observe the same lesion, AIW was marked and the coordinates of the lesion were recorded.

## Fluorescence activated cell sorting

Dissociated PDAC cells were incubated with an eFluor-conjugated anti-Epcam antibody (Thermo Fisher Scientific) for 30 min. Labeled cells were sorted on a FACS Aria II (BD Biosciences) on the basis of Dclk1 and Epcam expression using endogenous EGFP and eFluor-conjugated anti-Epcam antibody signals. FACS data were analyzed using FlowJo software (FlowJo, LLC, Ashland, OR) and FACS Diva software (version 8.0, BD Biosciences). The collected cells were embedded in growth factor-reduced Matrigel and cultured in culture medium supplemented with 10 µmol/l Y-27632 (Tocris Cookson, Bristol, UK).

## Microarray and gene enrichment analysis

RNA Integrity Numbers (RIN) were measured by Agilent 2200 TapeStation (Agilent Technologies, Santa Clara, CA). RIN of RNA samples used in microarray analysis were above 7. RNA samples were amplified, labeled, and hybridized to SurePrint G3 Mouse GE v2 8 × 60K Microarray (Agilent Technologies). Raw data were quantified and normalized by GeneSpring GX 14.5 software (Agilent Technologies). Genes whose signal was deemed invalid in all tissues by Signal Evaluation were excluded. Unnamed genes were also excluded. Signal data of overlapping probes were averaged. The heatmap was generated using R (Institute for Statistics and Mathematics) from differentially expressed genes with a significant difference, $p<0.01$ in gene expression. Pathway analysis and GO enrichment analysis were performed on DAVID 6.8 (Laboratory of Human Retrovirology and Immunoinformatics). Gene set enrichment analysis was performed by GSEA 3.0 software (Broad Institute) with 1000 gene-set permutations using the gene-ranking metric $t$-test with H collection: Hallmark gene sets and C2 collection: Curated gene sets in Molecular Signatures Database. Correlation was assessed by computing Pearson correlation coefficients.

## Xenotransplantation of PDAC cells

Dclk1$^+$ and Dclk1$^-$ mouse PDAC cells sorted by FACS were suspended in 100 µl of culture medium and injected subcutaneously into the flank of NOD/SCID mice. Tumor volumes were calculated according to the formula (length × width × height)/2.

## Quantification and statistical analysis

Data were presented as means ± SEM. The two-tailed Student's $t$-test was used to determine whether there was a significant difference between two groups. The chi-squared test was used to compare the formation rate of xenograft. p-Values less than 0.05 were considered statistically significant. $p<0.05$, $p<0.01$, and $p<0.001$ were represented with single, double, and triple asterisks, respectively.

## Acknowledgements

We thank Michiyuki Matsuda and Kyoto University Live Imaging Center for providing technical support of live imaging and Yuzo Kodama and all members of Fukuda, Seno, and Kodama laboratories for helpful suggestions.

## Additional information

### Funding

| Funder | Grant reference number | Author |
|---|---|---|
| Japan Society for the Promo- | Grants-in-Aid KAKENHI | Hiroshi Seno |

| | | |
|---|---|---|
| tion of Science | 26293173 | |
| Japan Society for the Promotion of Science | Grants-in-Aid KAKENHI 15H06334 | Takahisa Maruno |
| Japan Society for the Promotion of Science | Grants-in-Aid KAKENHI 16K15427 | Hiroshi Seno |
| Japan Society for the Promotion of Science | Grants-in-Aid KAKENHI 17H04157 | Hiroshi Seno |
| Japan Society for the Promotion of Science | Grants-in-Aid KAKENHI 16K09394 | Akihisa Fukuda |
| Japan Society for the Promotion of Science | Grants-in-Aid KAKENHI 19H03639 | Akihisa Fukuda |
| Japan Society for the Promotion of Science | Grants-in-Aid KAKENHI 20H03659 | Akihisa Fukuda |
| Japan Society for the Promotion of Science | Grants-in-AidKAKENHI 19K22619 | Akihisa Fukuda |
| Japan Agency for Medical Research and Development | 19cm0106142h0002 | Hiroshi Seno |
| Japan Agency for Medical Research and Development | 19cm6010022h0002 | Akihisa Fukuda |
| Japan Agency for Medical Research and Development | 20cm0106375h0001 | Akihisa Fukuda |
| Japan Agency for Medical Research and Development | 18cm0106142h0001 | Akihisa Fukuda |
| Japan Agency for Medical Research and Development | 20gm6010022h0003 | Akihisa Fukuda |
| Kobayashi Foundation for Cancer Research | | Hiroshi Seno |
| Naito Foundation | | Hiroshi Seno |
| Naito Foundation | 20829-1 | Dieter Saur |
| Takeda Science Foundation | 201749741 | Tsutomu Chiba |
| Uehara Memorial Foundation | 201720143 | Hiroshi Seno |
| Mochida Memorial Foundation for Medical and Pharmaceutical Research | 201356 | Akihisa Fukuda |
| Mochida Memorial Foundation for Medical and Pharmaceutical Research | 2017bvAg | Akihisa Fukuda |
| Mitsubishi Foundation | 281119 | Akihisa Fukuda |
| European Research Council | 648521 | Akihisa Fukuda |
| Deutsche Forschungsgemeinschaft | 1374/4-2 | Akihisa Fukuda |
| Princess Takamatsu Cancer Research Fund | 17-24924 | Dieter Saur |

The funders had no role in study design, data collection and interpretation, or the decision to submit the work for publication.

### Author contributions

Takahisa Maruno, Conceptualization, Data curation, Formal analysis, Supervision, Funding acquisition, Validation, Investigation, Visualization, Methodology, Writing - original draft, Project administration, Writing - review and editing; Akihisa Fukuda, Conceptualization, Data curation, Supervision, Funding acquisition, Investigation, Methodology, Project administration, Writing - review and editing; Norihiro Goto, Data curation, Supervision, Investigation, Methodology, Writing - review and editing; Motoyuki Tsuda, Supervision, Validation, Investigation, Methodology, Writing - review and

editing; Kozo Ikuta, Yukiko Hiramatsu, Validation, Investigation, Methodology, Writing - review and editing; Satoshi Ogawa, Yuki Nakanishi, Data curation, Formal analysis, Supervision, Validation, Investigation, Visualization, Methodology, Project administration, Writing - review and editing; Yuichi Yamaga, Validation, Investigation, Methodology; Takuto Yoshioka, Resources, Supervision, Validation, Investigation, Methodology, Writing - review and editing; Kyoichi Takaori, Resources, Supervision, Writing - review and editing; Shinji Uemoto, Resources, Supervision, Funding acquisition, Writing - review and editing; Dieter Saur, Tsutomu Chiba, Conceptualization, Resources, Data curation, Supervision, Funding acquisition, Project administration, Writing - review and editing; Hiroshi Seno, Conceptualization, Resources, Data curation, Supervision, Funding acquisition, Visualization, Methodology, Project administration, Writing - review and editing

### Author ORCIDs
Takahisa Maruno https://orcid.org/0000-0002-7060-4104
Akihisa Fukuda https://orcid.org/0000-0002-1940-596X

### Ethics
Human subjects: Surgically resected specimens of pancreatic cancer tissues were obtained from patients who had been admitted to Kyoto University Hospital. Written informed consent was obtained from all patients and study protocol (#G1200-1) was approved by Ethics Committee of Kyoto University Hospital.
Animal experimentation: All animal experiments were approved by the animal research committee of the Kyoto University and performed in accordance with Japanese government regulations. All surgery was performed under Isoflurane anesthesia, and every effort was made to minimize suffering.

### Decision letter and Author response
Decision letter https://doi.org/10.7554/eLife.55117.sa1
Author response https://doi.org/10.7554/eLife.55117.sa2

# Additional files

### Supplementary files
• Supplementary file 1. GO enrichment analysis up to 100 Go terms on DAVID, GO Biological process using 2171 genes significantly (p<0.01) highly expressed in Dclk1$^+$ PDAC cells.

• Supplementary file 2. Pathway analysis on DAVID, KEGG Pathway using 2171 genes significantly (p<0.01) highly expressed in Dclk1$^+$ PDAC cells.

• Supplementary file 3. GO analysis (GO Biological Process) and pathway analysis (KEGG Pathway) on genes that were significantly highly expressed (p<0.05) in the DCLK1 high expression group up to 50 Go terms or pathways.

• Transparent reporting form

### Data availability
Microarray data have been deposited in GEO under accession codes GSE139167.

The following dataset was generated:

| Author(s) | Year | Dataset title | Dataset URL | Database and Identifier |
|---|---|---|---|---|
| Maruno T, Seno H | 2019 | Gene expression profiles of Dclk1+ and Dclk1- PDAC cells | https://www.ncbi.nlm.nih.gov/geo/query/acc.cgi?acc=GSE139167 | NCBI Gene Expression Omnibus, GSE139167 |

The following previously published datasets were used:

| Author(s) | Year | Dataset title | Dataset URL | Database and Identifier |
|---|---|---|---|---|
| Pei H, Li L, Fridley BL, Jenkins G, Kalari KR, Lingle W, Gloria PM, Lou Z, Wang L | 2009 | Expression data from Mayo Clinic Pancreatic Tumor and Normal samples | https://www.ncbi.nlm.nih.gov/geo/query/acc.cgi?acc=GSE16515 | NCBI Gene Expression Omnibus, GSE16515 |
| Donahue TR, Tran LM, Hill R, Li Y, Kovochich A, Calvopina JH, Patel SG, Wu N, Hindoyan A, Farrell JJ, Li X, Dawson DW, Wu H | 2011 | Integrative Survival-Based Molecular Profiling of Human Pancreatic Cancer | https://www.ncbi.nlm.nih.gov/geo/query/acc.cgi?acc=GSE32676 | NCBI Gene Expression Omnibus, GSE32676 |
| Biankin AV, Waddell N, Kassahn K, Gingras M, Johns AL, Miller D, Wilson P, Patch A, Wu J, Chang DK, Cowley MJ, Gardiner B, Song S, Harliwong I, Idrisoglu S, Nourse C, Nourbakhsh E, Manning S, Wani S, Gongora M, Pajic M, Scarlett CJ, Gill AJ, Musgrove EA, Sutherland RL, Pinho AV, Rooman I, Anderson M, Holmes O, Leonard C, Taylor D, Wood S, Xu C, Nones K, Fink L, Christ A, Bruxner T, Cloonan N, Kolle G, Newell F, Pinese M, Humphris JL, Kaplan W, Jones MD, Colvin EK, Nagrial AM, Humphrey ES, Chou A, Chin VT, Chantrill LA, Samra JS, Kench JG, Pettit J, Daly RJ, Merrett ND, Toon C, Epari K, Nguyen NQ, Barbour A, Zeps N, Kakkar N, Zhao F, Wu YQ, Wang M, Muzny DM, Fisher WE, Brunicardi FC, Hodges SE, Drummond J, Chang K, Han Y, Lewis L, Dinh H, Buhay C, Muthuswamy L, Beck T, Timms L, Sam M, Begley K, Brown A, Pai D, Panchal A, Buchner N, De Borja R, Denroche R, Yung C, Serra S, Onetto N, Mukhopadhyay D, Tsao M, Shaw | 2012 | ICGC Pancreas: Genomic analysis reveals roles for chromatin modification and axonguidance in pancreatic cancer | https://www.ncbi.nlm.nih.gov/geo/query/acc.cgi?acc=GSE36924 | NCBI Gene Expression Omnibus, GSE36924 |

PA, Petersen G, Gallinger S, Stein LD, Hruban RH, Maitra A, Iacobuzio-Donahue CA, Schulick RD, Wolfgang CL, Morgan R, Lawlor RT, Beghelli S, Corbo V, Scardoni M, Bassi C, Tempero MA, Mann KM, Jenkins NA, Perez-Mancera PA, Adams DJ, Largaespada DA, Wessels LF, Rust AG, Tuveson DA, Copeland NG, Hudson TJ, Scarpa A, Eshleman JR, Wheeler DA, Pearson JV, McPherson JD, Gibbs RA, Grimmond SM

| The Cancer Genome Atlas Research Network | 2017 | Pancreatic Adenocarcinoma | https://www.cbioportal.org/study/summary?id=paad_tcga_pan_can_atlas_2018 | The Cancer Genome Atlas, paad_tcga_pan_can_atlas_2018 |
|---|---|---|---|---|

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
