## [Decision Letter]

**Acceptance summary:**

Your manuscript was found by reviewers to be significant in demonstrating that DCLK1^+^ cells express stem cell markers and contribute to PDAC tumor and metastasis over time. In particular, your imaging provides compelling evidence that these cells should be targeted in PDAC.

**Decision letter after peer review:**

Thank you for submitting your article "Visualization of stem cell activity in pancreatic cancer expansion by direct lineage tracing with live imaging" for consideration by *eLife*. Your article has been reviewed by three peer reviewers, one of whom is a member of our Board of Reviewing Editors, and the evaluation has been overseen by Richard White as the Senior Editor. The reviewers have opted to remain anonymous.

The reviewers have discussed the reviews with one another and the Reviewing Editor has drafted this decision to help you prepare a revised submission.

All reviewers were impressed by aspects of the data but also raised a limited number of concerns. We would be grateful if you could address these concerns and in particular provide additional data to address the concern of reviewer #3 about the leakiness of Cre expression and whether you are confusing DCLK^+^ cells with their progeny. In order to be more favorably reviewed upon revision, it will be critical to address all of the points raised by the reviewers, which are listed below.

Reviewer #1:

DCLK has been previously reported to mark cancer initiating cells in PDAC (Bailey et al., 2014) and previous lineages tracing studies have supported this contention by showing that progeny of DCLK^+^ cells are expanded in response to pancreatitis and in PanIN lesions (Westphalen et al., 2016). The current studies extended these lineage tracing analyses to show DCLK^+^ cell progeny in PDAC and in liver metastases and show that mouse and human DCLK^+^ sorted cells uniquely give rise to PDAC in transplanted mice and that these cells express high levels of stem cell markers, EMT and invasiveness markers.

Overall the data are convincing and the figures clear and well presented. The authors should address the following points:

1) There is some concern about the small numbers of mice used in the study (an n = 3 for most points). Nor is it clear from the figure legends how many lesions per mouse were examined. Do the figures show "cherry picked" EGFP^+^ lesions, or are these representative of all positive lesions? The authors need to clarify this.

2) Additionally, the authors report that 500 DCLK^+^ cells do not form tumors in transplanted nude mice which raises the concern that DCLK marks a population larger than just stem cells. Can the authors comment on this?

Reviewer #2:

The manuscript by Maruno and others presents a novel take on identifying DCLK1^+^ cells as cancer stem cells by employing lineage tracing and in vivo imaging to demonstrate that DCLK1^+^ cells give rise to DCLK1^-^ PanIN and PDAC cells. Overall, the work is intriguing and well-done. The in vivo imaging provides particularly convincing evidence that DCLK1^+^ cells give rise to the bulk of DCLK1^-^ PanIN and PDAC cells.

A few major points are as follows:

1) The cancer stem cell marker data is weak. Since Epcam, Cd44, and Cd24 are all cancer stem cell markers, they are not expected to be present in all PanIN and PDAC cells. This is in contrast to the staining shown in Figure 1I and quantification in Figures 1J, K. Additionally, Cd24 appears present in all PanIN cells, but the quantification indicates that less than 20% of PanIN cells are Cd24^+^. Additional cancer stem cell markers should be used or alternative antibodies for Epcam, Cd44, and Cd24 should be used.

2) Explicitly showing that EGFP^+^ cells are DCLK1^+^ before addition of tamoxifen is an important control. The figures in which this is relevant are noted below:

a) In Figures 2D and 3D, the Day 0 data should be shown in conjunction with DCLK1 staining to show that the EGFP^+^ cells are indeed DCLK1^+^.

b) In Supplementary Figure 1C, immunofluorescent staining for DCLK1 should be shown in conjunction with EGFP and Tomato to show that there is a DCLK1^+^ cell present giving rise to the EGFP^+^ cells. Additional overlay with DAPI would enhance image interpretation.

c) In Figure 4E, immunofluorescent staining for DCLK1 should be shown in conjunction with EGFP and Tomato to show that there is a DCLK1^+^ cell present giving rise to the EGFP^+^ cells. Additional overlay with DAPI would enhance image interpretation.

Reviewer #3:

In this manuscript, Maruno et al. set out to test the hypothesis that DCLK1^+^ cells within pancreatic neoplasia and cancer act as cancer stem cells, using advanced lineage tracing experiments and analysis. Clearly the manuscript represents a tremendous amount of work, but unfortunately the central premise isn't well supported largely because the DCLK1CreER recombinase mouse is not characterized sufficiently to draw the conclusions therein. Even with proper characterization, some of the issues that arise leave significant doubt that the data could be so simply interpreted.

1) The DCLK-CreER in the manuscript is demonstrably leaky, making assumptions about accurate lineage tracing under most conditions questionable. Is this leakiness specific to DCLK1 positive cells? If so, what percentage of cells are being labeled? What about in the normal pancreas and in Kras and Kras;p53 mice prior to transformation? If the leakiness is specific to DCLK1^+^ cells, why do those cells not appear to expand (i.e. they remain isolated in the figures shown).

2) There is no demonstration that the CreER driven recombination is specific to DCLK1 positive cells after tamoxifen treatment. Shorter term analysis after tamoxifen treatment co-registered with DCLK1 staining is required. The authors should keep in mind that tamoxifen stays within the animal's system for several days when designing these experiments.

3) Westphalen et al. observe DCLK1CreER recombination in a subpopulation of normal acinar and duct cells, which still exist in the background of the authors' model systems. There's no question that acinar and/or duct cells will prove to be the source of tumor, therefore recombination within either compartment will confound interpretation. How is this possibility being distinguished compared to the occasional labeling of a cell that has already been transformed? (These first three points are absolutely critical issues, illustrating that the CreER system has not been characterized sufficiently for the authors to draw their primary conclusions.)

4) The authors show that DCLK1 positive cell enriched human cancers and DCLK1 positive cell-derived daughter cells in the mouse express higher levels of EMT markers. The authors (and others) show that cells that *maintain* DCLK1 expression in pancreatic cancers are very rare. Thus, there is a logical inconsistency with these data in that if the central premise where DCLK1-positive cells give rise DCLK1-negative progeny with EMT characteristics, then the EMT signature in the human data sets would no longer track with DCLK1 expression itself. In other words, the authors seem to be conflating DCLK1 cell progeny with DCLK1 cells themselves.

---

## [Author Response]

All reviewers were impressed by aspects of the data but also raised a limited number of concerns. We would be grateful if you could address these concerns and in particular provide additional data to address the concern of reviewer #3 about the leakiness of Cre expression and whether you are confusing DCLK^+^ cells with their progeny. In order to be more favorably reviewed upon revision, it will be critical to address all of the points raised by the reviewers, which are listed below.

We thank the editors and reviewers for the constructive and helpful comments. As described in detail below, we have followed their suggestions and performed the requested experiments to strengthen the impact of our findings. We strongly believe that we have faithfully and thoroughly responded to all of the points raised by the reviewers. Of note, all additional findings confirmed the conclusions presented in the original manuscript and we feel that the revisions strongly support our conclusion that we provided direct functional evidence for the stem cell activity of Dclk1^+^ cells in vivo, revealing the essential roles of Dclk1^+^ cells in expansion of pancreatic neoplasia in all progressive stages. Our responses are written in blue font.

Reviewer #1:[…] Overall the data are convincing and the figures clear and well presented. The authors should address the following points:1) There is some concern about the small numbers of mice used in the study (an n = 3 for most points). Nor is it clear from the figure legends how many lesions per mouse were examined. Do the figures show "cherry picked" EGFP^+^ lesions, or are these representative of all positive lesions? The authors need to clarify this.

We thank the reviewer for the constructive comment. In this revision, we have extensively increased the numbers of mice analyzed. The number of mice has been more than 5 for each analysis. The numbers of lesions per mouse examined have now been described in the figure legends (Figures 2, 3 and 4). EGFP^+^ lesions are representative of all positive lesions. We have presented representative images in the revised Figures 2F, 3F, H, and 4G.

2) Additionally, the authors report that 500 DCLK^+^ cells do not form tumors in transplanted nude mice which raises the concern that DCLK marks a population larger than just stem cells. Can the authors comment on this?

In this revision, we have re-performed xenograft experiments with more conditions including injection with 100 or 500 Dclk1^+^ cells as well as Dclk1^-^ cells. The results showed that tumors were formed from either 100 or 500 Dclk1^+^ PDAC cells. In contrast, tumors were never formed even when 10,000 Dclk1^-^ PDAC cells were transplanted (Figure 6H). Therefore, our data indicate that Dclk1 marks PDAC stem cells with a tumor-forming ability in vivo.

Reviewer #2:[…] A few major points are as follows:1) The cancer stem cell marker data is weak. Since Epcam, Cd44, and Cd24 are all cancer stem cell markers, they are not expected to be present in all PanIN and PDAC cells. This is in contrast to the staining shown in Figure 1I and quantification in Figures 1J, K. Additionally, Cd24 appears present in all PanIN cells, but the quantification indicates that less than 20% of PanIN cells are Cd24^+^. Additional cancer stem cell markers should be used or alternative antibodies for Epcam, Cd44, and Cd24 should be used.

We thank the reviewer for the constructive comment. Regarding staining for Epcam, Cd44 and Cd24, we have reperformed immunostaining using different kind of antibodies in this revision. The results showed that the positive ratio of Epcam and Cd44 was approximately 70% in both mouse PanIN and PDAC cells and that the positive ratio of Cd24 was about 17% in both PanIN and PDAC cells. Although Epcam, Cd44 and Cd24 have been shown to be cancer stem cell markers of PDAC, the single expression of Epcam, Cd44 or Cd24 alone is not restricted to cancer stem cells in PDAC (Kure et al., 2012). That accounts for relatively high percentage of their expression in PanIN and PDAC cells. The new images and results have been presented in the revised Figure 1I, J, K.

2) Explicitly showing that EGFP^+^ cells are DCLK1^+^ before addition of tamoxifen is an important control. The figures in which this is relevant are noted below:a) In Figures 2D and 3D, the Day 0 data should be shown in conjunction with DCLK1 staining to show that the EGFP^+^ cells are indeed DCLK1^+^.

We thank the reviewer for the constructive comment. We have added images of immunostaining for Dclk1 and GFP before tamoxifen administration, showing that EGFP^+^ cells were indeed DCLK1 positive before tamoxifen administration. These new data have been presented in the revised Figures 2C and 3C.

b) In Supplementary Figure 1C, immunofluorescent staining for DCLK1 should be shown in conjunction with EGFP and Tomato to show that there is a DCLK1^+^ cell present giving rise to the EGFP^+^ cells. Additional overlay with DAPI would enhance image interpretation.

We thank the reviewer for the comment. In this revision, we acquired the fluorescent images of EGFP and Tomato in conjunction with nuclear staining for Hoechst in the lineage tracing analysis of PDAC organoids. They clearly show that there is a DCLK1^+^ cell present giving rise to the EGFP^+^ cells. The new images have been presented in the revised Figure 3—figure supplement 2C.

c) In Figure 4E, immunofluorescent staining for DCLK1 should be shown in conjunction with EGFP and Tomato to show that there is a DCLK1^+^ cell present giving rise to the EGFP^+^ cells. Additional overlay with DAPI would enhance image interpretation.

We thank the reviewer for the constructive comment. In this revision, we acquired the fluorescent images of EGFP and Tomato in conjunction with nuclear Hoechst staining in the lineage tracing analysis of liver tumors. They clearly show that there is a DCLK1^+^ cell present giving rise to the EGFP^+^ cells. The new images have been presented in the revised Figure 4F and G.

Reviewer #3:In this manuscript, Maruno et al. set out to test the hypothesis that DCLK1^+^ cells within pancreatic neoplasia and cancer act as cancer stem cells, using advanced lineage tracing experiments and analysis. Clearly the manuscript represents a tremendous amount of work, but unfortunately the central premise isn't well supported largely because the DCLK1CreER recombinase mouse is not characterized sufficiently to draw the conclusions therein. Even with proper characterization, some of the issues that arise leave significant doubt that the data could be so simply interpreted.1) The DCLK-CreER in the manuscript is demonstrably leaky, making assumptions about accurate lineage tracing under most conditions questionable. Is this leakiness specific to DCLK1 positive cells? If so, what percentage of cells are being labeled? What about in the normal pancreas and in Kras and Kras;p53 mice prior to transformation? If the leakiness is specific to DCLK1^+^ cells, why do those cells not appear to expand (i.e. they remain isolated in the figures shown).

We thank the reviewer for the insightful comment. In this study, we used the *Dclk1^creERT2-IRES-EGFP^* knock-in mice which we originally created, although no analysis had been performed regarding Cre leakiness in pancreas. According to the reviewer’s suggestion, we extensively analyzed whether there is a potential leakiness of CreER expression in this *Dclk1^creERT2-IRES-EGFP^* knock-in mice in the revised manuscript. For this, we applied *Dclk1^creERT2-IRES-EGFP/+^* mice without tamoxifen administration, in which one allele harbors wild-type *Dclk1* while the other harbors *creERT2-IRES-EGFP* transgene cassette under *Dclk1* promoter region. If there was a leakiness of *creERT2-IRES-EGFP*, we would observe inconsistent expression pattern between Dclk1 and creERT2-IRES-EGFP. Importantly, double immunofluorescence revealed that Dclk1 staining was completely consistent with that of EGFP in normal pancreas and pancreatic epithelium irrespective of Kras and/or p53 mutation status. For this evaluation, we analyzed 536 cells in normal pancreas, 604 cells in pancreatic epithelium with Kras mutation and 685 cells in pancreatic epithelium cells with both Kras mutation and p53 deletion. Therefore, we concluded that the *Dclk1^creERT2-IRES-EGFP^* mice we used in this study had no leakiness of CreER expression at all prior to transformation. These data have been shown in the revised Figure 3—figure supplement 1A.

2) There is no demonstration that the CreER driven recombination is specific to DCLK1 positive cells after tamoxifen treatment. Shorter term analysis after tamoxifen treatment co-registered with DCLK1 staining is required. The authors should keep in mind that tamoxifen stays within the animal's system for several days when designing these experiments.

We thank the reviewer for the constructive comment. As the reviewer suggested, in this revision, we investigated whether CreER driven recombination occurred in Dclk1-negative cells in *Dclk1^creERT2-IRES-EGFP^* mice after tamoxifen administration. Given that tamoxifen remains in the mouse body for several days, we extensively analyzed the number of GFP-positive cells in Dclk1-negative cells 1 and 3 days after tamoxifen administration. The results demonstrated that GFP-positive cells in Dclk1-negative cells were hardly seen in pancreatic epithelium of the mice with no gene alteration, with Kras single mutation or with both Kras mutation and p53 deletion on day 1 and day 3 after tamoxifen administration. Indeed, the proportion of GFP^+^ cells among Dclk1^-^ cells was extremely low (almost zero); 0.0070 ± 0.0043 % at day 1, 0.0061 ± 0.0038 % at day 3 in normal pancreatic epithelium, 0.0069 ± 0.0042 % at day 1, 0.0057 ± 0.0051 % at day 3 in pancreatic epithelium with Kras mutation, and 0.0061 ± 0.0067 % at day 1, 0.0061 ± 0.0061 % at day 3 in pancreatic epithelium with Kras mutation and p53 deletion. We analyzed more than 5 mice per genotype for this analysis. Therefore, given the extremely low percentage of Cre recombination in Dclk1 negative cells upon tamoxifen administration, we strongly believe that it is acceptable to conclude that CreER driven recombination is specific to Dclk1 positive cells after tamoxifen treatment in *Dclk1^creERT2-IRES-EGFP^* mice in this study. These new data have been presented in the revised Figure 3—figure supplement 1B.

3) Westphalen et al. observe DCLK1CreER recombination in a subpopulation of normal acinar and duct cells, which still exist in the background of the authors' model systems. There's no question that acinar and/or duct cells will prove to be the source of tumor, therefore recombination within either compartment will confound interpretation. How is this possibility being distinguished compared to the occasional labeling of a cell that has already been transformed? (These first three points are absolutely critical issues, illustrating that the CreER system has not been characterized sufficiently for the authors to draw their primary conclusions.)

We thank the reviewer for the insightful comment. In this study, we established a novel system to trace the dynamics of CSCs by utilizing an inducible dual-recombinase system that combined flippase-*FRT* and Cre-*loxP* recombinations in mice. As the reviewer pointed out, it is not possible to completely rule out the possibility that Dclk1-positive acinar/duct cells give rise to PanIN/PDAC cells during lineage tracing analysis after tamoxifen induction. Also, it is not possible to precisely and technically determine whether all Dclk1-lineage labeled PanIN/PDAC cells were originally PanIN/PDAC cells when tamoxifen was administered. However, we performed live imaging techniques to observe the expansion of tumor cells within the same tumors in the same individual mice. Taking advantage of this novel method, we unequivocally demonstrated that Dclk1^+^ tumor cells supply descendant cells in both established PanIN and PDAC lesions in live mouse during the observation period. Moreover, regarding lineage tracing analysis of PDAC, it is extremely unlikely that acinar/duct cells give rise to PDAC cells so quickly (within 14 days; Figure 3H and I). Furthermore, we performed lineage tracing analysis of liver metastasis and PDAC organoids in this study. In the setting of liver metastasis and PDAC organoids, there were no involvements of acinar/duct cells at all. Therefore, our lineage tracing data of liver metastasis and PDAC organoids clearly demonstrated that Dclk1^+^ PDAC cells give rise to descendant PDAC cells.

4) The authors show that DCLK1 positive cell enriched human cancers and DCLK1 positive cell-derived daughter cells in the mouse express higher levels of EMT markers. The authors (and others) show that cells that maintain DCLK1 expression in pancreatic cancers are very rare. Thus, there is a logical inconsistency with these data in that if the central premise where DCLK1-positive cells give rise DCLK1-negative progeny with EMT characteristics, then the EMT signature in the human data sets would no longer track with DCLK1 expression itself. In other words, the authors seem to be conflating DCLK1 cell progeny with DCLK1 cells themselves.

We apologize for confusing the reviewer. The *Dclk1 ^CreERT2-IRES-EGFP /+^; Pdx1-Flp; Kras^FSF-G12D/+^; Trp53^frt/frt^* (*DKPF*) mice which we used in the microarray analysis did not contain the Rosa-mTmG reporter allele and they were not treated with tamoxifen at all. Therefore, as described above in the response to the point 1), the GFP-positive cells were not DCLK1 positive cell-derived daughter cells but 100% absolutely Dclk1-expressing cells themselves. In other words, we did not conflate DCLK1 cell progeny with DCLK1 cells themselves in this analysis. Therefore, our conclusion obtained from FACS and microarrays data analysis was that Dclk1-positive cells have higher EMT activity compared to Dclk1-negative cells., and we strongly believe that there is no logical contradiction or inconsistency regarding our mouse and human data.